# Mechanism and structural dynamics of sulfur transfer during de novo [2Fe-2S] cluster assembly on ISCU2

Vinzent Schulz[1,2,8], Ralf Steinhilper[3,8], Jonathan Oltmanns[4,8], Sven-A. Freibert[1,2,7], Nils Krapoth[1,2], Uwe Linne[5], Sonja Welsch[6], Maren H. Hoock[4], Volker Schünemann[4], Bonnie J. Murphy[3] ✉ & Roland Lill[1,2] ✉

Maturation of iron-sulfur proteins in eukaryotes is initiated in mitochondria by the core iron-sulfur cluster assembly (ISC) complex, consisting of the cysteine desulfurase sub-complex NFS1-ISD11-ACP1, the scaffold protein ISCU2, the electron donor ferredoxin FDX2, and frataxin, a protein dysfunctional in Friedreich's ataxia. The core ISC complex synthesizes [2Fe-2S] clusters de novo from Fe and a persulfide (SSH) bound at conserved cluster assembly site residues. Here, we elucidate the poorly understood Fe-dependent mechanism of persulfide transfer from cysteine desulfurase NFS1 to ISCU2. High-resolution cryo-EM structures obtained from anaerobically prepared samples provide snapshots that both visualize different stages of persulfide transfer from Cys381[NFS1] to Cys138[ISCU2] and clarify the molecular role of frataxin in optimally positioning assembly site residues for fast sulfur transfer. Biochemical analyses assign ISCU2 residues essential for sulfur transfer, and reveal that Cys138[ISCU2] rapidly receives the persulfide without a detectable intermediate. Mössbauer spectroscopy assessing the Fe coordination of various sulfur transfer intermediates shows a dynamic equilibrium between pre- and post-sulfur-transfer states shifted by frataxin. Collectively, our study defines crucial mechanistic stages of physiological [2Fe-2S] cluster assembly and clarifies frataxin's molecular role in this fundamental process.

Iron-sulfur (Fe/S) clusters are ancient inorganic protein cofactors present in almost all organisms, and are involved in numerous biological processes including respiration, metabolism, DNA replication and repair, translation, regulation, and antiviral defense[1–6]. Despite the chemical simplicity of Fe/S clusters, their biosynthesis and insertion into apoproteins requires complex proteinaceous machineries[7–10]. In eukaryotes, the mitochondrial Fe/S cluster assembly (ISC) system synthesizes Fe/S clusters de novo and inserts them into mitochondrial target apoproteins. Genetic mutations in the human ISC genes impair this process and cause severe metabolic, neurological and

[1]Institut für Zytobiologie, Philipps-Universität Marburg, Karl-von-Frisch-Str. 14, 35032 Marburg, Germany. [2]Zentrum für Synthetische Mikrobiologie SynMikro, Karl-von-Frisch-Str. 14, 35032 Marburg, Germany. [3]Redox and Metalloprotein Research Group, Max Planck Institute of Biophysics, Max-von-Laue-Str. 3, 60438 Frankfurt am Main, Germany. [4]Department of Physics, Biophysics and Medical Physics, University of Kaiserslautern-Landau, Erwin-Schrödinger-Str. 46, 67663 Kaiserslautern, Germany. [5]Mass Spectrometry Facility of the Department of Chemistry, Philipps-Universität Marburg, Hans-Meerwein-Str. 4, 35032 Marburg, Germany. [6]Central Electron Microscopy Facility, Max Planck Institute of Biophysics, Max-von-Laue-Str. 3, 60438 Frankfurt am Main, Germany. [7]Present address: Steinmühle—Schule & Internat, Steinmühlenweg 21, 35043 Marburg, Germany. [8]These authors contributed equally: Vinzent Schulz, Ralf Steinhilper, Jonathan Oltmanns. ✉e-mail: bonnie.murphy@biophys.mpg.de; lill@staff.uni-marburg.de

hematological diseases, often with fatal outcomes, the most prevalent being Friedreich's ataxia with mutations in the frataxin (FXN) gene[9–13].

Mitochondrial Fe/S protein biogenesis begins with the de novo assembly of a [2Fe-2S] cluster from $Fe^{2+}$ and Cys-derived persulfide (SSH) on the scaffold protein ISCU2[14]. Assembly requires the multi-protein 'core ISC complex' consisting of ISCU2, the cysteine desulfurase sub-complex NFS1-ISD11-ACP1, FXN, and the ferredoxin

FDX2[15–20] (Fig. 1a). The assembled [2Fe-2S] cluster is then released from ISCU2 by a dedicated HSP70 chaperone system and transferred via numerous ISC trafficking and targeting factors toward the mitochondrial client Fe/S apoproteins[8,21–24].

Despite a number of biochemical and structural studies of the core ISC complex, the biochemical mechanism of de novo [2Fe-2S] cluster assembly on ISCU2 is poorly understood at the molecular level.

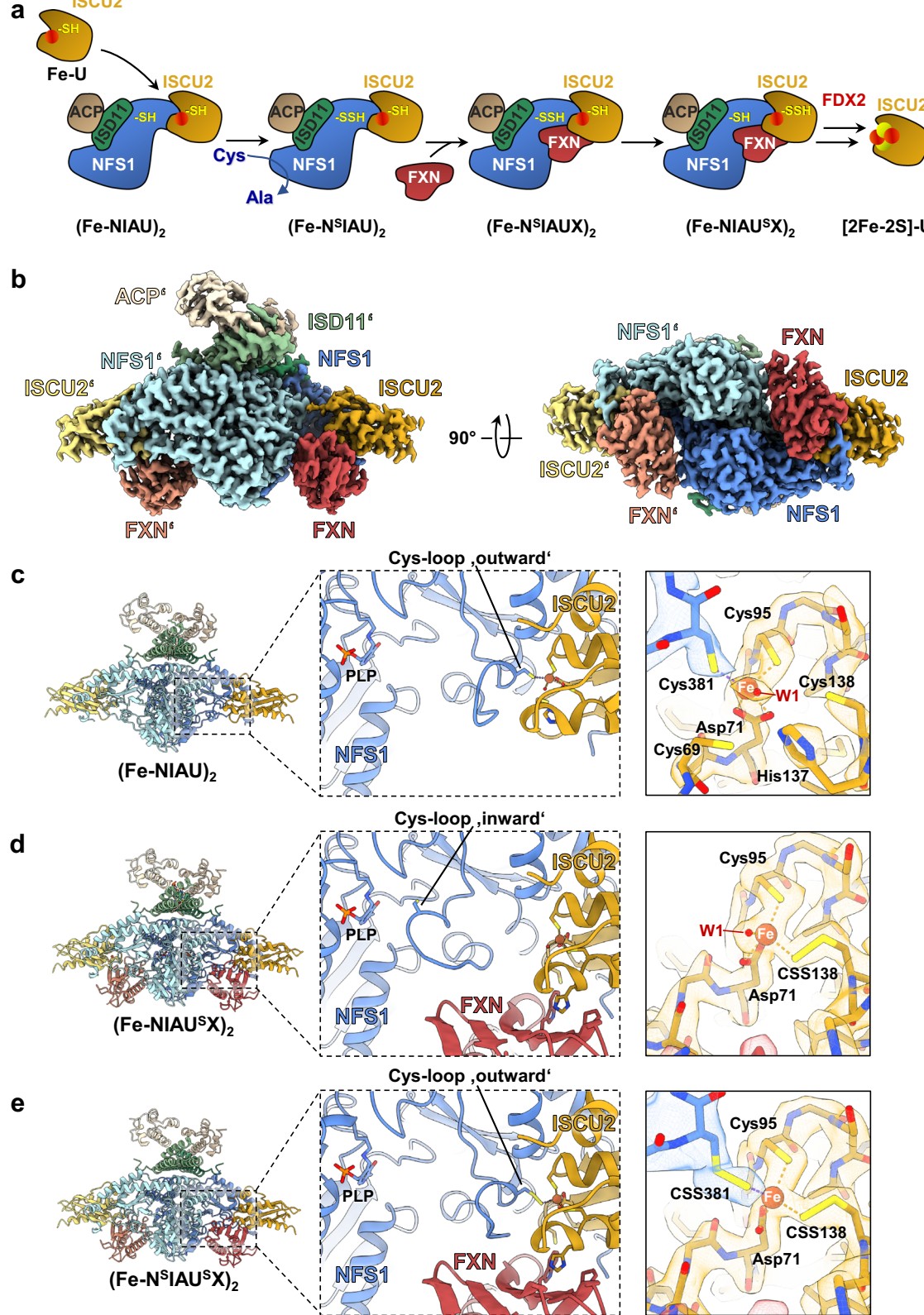

**Fig. 1 | Cryo-EM structures elucidate various stages of sulfur transfer from NFS1 to ISCU2 during de novo [2Fe-2S] cluster formation. a** Model for the synthesis of a [2Fe-2S] cluster on the ISCU2 scaffold protein by the core ISC complex in various steps. Free ISCU2 (Fe-U) with $Fe^{2+}$ (red circle) bound at its assembly site residues associates with the heterodimeric cysteine desulfurase complex NFS1-ISD11-ACP1 ($(NIA)_2$; only one half of the dimer is shown) to form the $(Fe-NIAU)_2$ complex. Addition of Cys enables rapid persulfidation of NFS1 forming the persulfide-containing $(Fe-N^SIAU)_2$ complex and Ala. FXN (X) addition stimulates sulfur transfer from NFS1 to a Cys residue at the ISCU2 assembly site creating the $(Fe-NIAU^SX)_2$ intermediate. FDX2-mediated persulfide reduction and interaction of two ISCU2 of different $(Fe-NIAU^SX)_2$ complexes leads to the formation of holo-ISCU2 with a bound [2Fe-2S] cluster. **b** 2.4 Å consensus cryo-EM map (C2 symmetry applied) of the $(Fe-NIAUX)_2$ complex during persulfide transfer, segmented and colored by subunit. Unless indicated otherwise, subunit coloring is consistent throughout the manuscript. Symmetry expansion and subsequent focused 3D classification allowed further separation of the $(Fe-NIAU)_2$ (**c**), $(Fe-NIAU^SX)_2$ (**d**), and $(Fe-N^SIAU^SX)_2$ (**e**) reaction intermediates (left). Middle: the cartoon representation shows a detailed view of the NFS1-ISCU2 interface with the different locations of the Cys loop of NFS1, either in an outward position with Cys381[NFS1] coordinating the $Fe^{2+}$ at the ISCU2 assembly site (**c, e**), or in an inward orientation (**d**) with Cys381[NFS1] close to the PLP cofactor of NFS1. Right: the close-up of the respective ISCU2 assembly site regions (with or without Cys381[NFS1]) depicts the different modes of $Fe^{2+}$ coordination in these sulfur transfer intermediates. The cryo-EM maps reveal clear densities for a persulfide (CSS) on Cys381[NFS1] and/or Cys138[ISCU2] (**d, e**).

**Table 1 | ISCU2-bound metal to ligand and sulfur to sulfur distances in various structures of free ISCU2 or core ISC complexes**

| Method | Solution NMR | X-ray | X-ray | Cryo-EM | Cryo-EM | Cryo-EM | Cryo-EM |
|---|---|---|---|---|---|---|---|
| Resolution | – | 3.32 Å | 3.32 Å | 2.75 Å | 3.20 Å | 2.58 Å | 2.49 Å |
| Reference | – | Boniecki (2017)[15] | Boniecki (2017)[15] | This work | Fox (2019)[17] | This work | This work |
| Organism | *Mus musculus* | *Homo sapiens* | *Homo sapiens* | *Homo sapiens* | *Homo sapiens* | *Homo sapiens* | *Homo sapiens* |
| PDB code | 1WFZ | 5WLW | 5WLW | 8PKA | 6NZU | 8PK9 | 8PK8 |
| Structure | Zn-U | $(Zn-NIAU)_2$ (Zn 201.D) | $(Zn-NIAU)_2$ (Zn 201.H) | $(Fe-NIAU)_2$ | $(Zn-NIAUX)_2$ | $(Fe-N^SIAU^SX)_2$ | $(Fe-NIAU^SX)_2$ |
| C69 | 2.3# | 6.1 | 4.8 | 3.7 | 10.0 | 9.6 | 9.7 |
| D71 OD1 | 3.5 | 3.1 | 2.9# | 2.1# | 2.3# | 2.1# | 2.0# |
| D71 OD2 | 2.1# | 2.4# | 2.4# | 2.2# | 2.3# | 2.1# | 2.1# |
| C95 | 2.3# | 2.8# | 2.9# | 2.3# | 2.8a | 2.4# | 2.3# |
| H137 NE2 | 2.1# | 2.8# | 3.2 | 4.8 | 10.5 | 9.8 | 9.9 |
| C138 | 3.6 | 3.9 | 3.9 | 4.7 | 2.9# | 2.3# | 2.3# |
| C381 (NFS1) | – | 2.9# | 2.8# | 2.4# | 12.3 | 2.3# | 20.9 |
| Water | – | – | – | 2.0# | – | – | 2.0# |
| | | | | | | | |
| C381-C69 | | 4.2 | 4.8 | 4.3 | | 9.6 | |
| C381-C95 | | 5.2 | 4.8 | 3.8 | | 3.4 | |
| C381-C138 | | 6.3 | 6.5 | 6.5 | | 3.6 | |

The Zn or Fe distances (in Å) to the indicated ligands are given in the top part, and distances smaller than 3 Å are marked by #. The bottom part shows the distances between the Cys381[NFS1] sulfur and indicated Cys[ISCU2] sulfurs.

aThe resolution of the map obtained in this study results in some ambiguity in sidechain positioning. Manual re-refinement of the model to the published density resulted in a measured distance of 2.3 Å.

So far, structural insights are restricted to complexes with either no metal or inhibitory Zn bound to ISCU2 rather than the physiological Fe. No structures of reaction intermediates are available to date. Biochemical studies revealed that initially the NFS1-ISD11-ACP1 heterodimer (hereafter termed $(NIA)_2$) uses free Cys to generate alanine and a persulfide bound to the conserved Cys381[NFS1] [25,26] (Fig. 1a). This residue is located on a flexible loop (Cys loop) that moves toward the $Fe^{2+}$ binding site of ISCU2 (denoted as U)[19], thereby forming the $(Fe-N^SIAU)_2$ complex (S indicates persulfidation of the preceding ISC component). The $(NIAU)_2$ complex was shown to have increased affinity for FXN (X)[16,27–29]. FXN binding to generate the intermediate $(Fe-N^SIAUX)_2$ complex strongly increases the rate of persulfide transfer from NFS1 to ISCU2 (yielding $(Fe-NIAU^SX)_2$) (Fig. 1a), implying a dedicated physiological role of FXN in this step, but FXN's precise mechanistic function remains unclear[17,25,26,30]. Studies performed aerobically suggested Cys69[ISCU2] and Cys138[ISCU2] as potential primary residues accepting the sulfur from NFS1, and Cys138[ISCU2] as the final acceptor[25]. In the human $(Zn-NIAU)_2$ X-ray structure[15], the latter residue is furthest from the sulfur-donating Cys381[NFS1] (distances for ligand-metal coordination and for Cys-Cys pairs in various ISC complex structures are compiled in Table 1). In the $(Zn-NIAUX)_2$ electron cryo-microscopy (cryo-EM) structure[17], a rather different arrangement was observed at the metal

site, with Cys381[NFS1] being far away from the ISCU2-bound Zn and its coordinating residues, hence providing no clues on the mechanism of sulfur transfer. Clearly, structural information for the Fe-containing ISC complex intermediates is needed. Notably, in the related *Bacillus subtilis* SufS-SufU sulfur transfer system, Cys41[SufU] (corresponding to Cys69[ISCU2]) serves as the sulfur-accepting residue indicating a conspicuous difference for the ISC and SUF systems[31].

Following sulfur transfer, the persulfide bound to ISCU2 in the $(Fe-NIAU^SX)_2$ complex is reduced to form sulfide ($S^{2-}$) by FDX2, which receives electrons from the NADPH-dependent ferredoxin reductase (FDXR)[14,20,30,32]. The reduction step generates a transient [1Fe-1S] intermediate that is finally fused to a [2Fe-2S] cluster on ISCU2 by dimerization facilitated by the hydrophobic interaction of the conserved N-terminal Tyr35 residues of two [1Fe-1S]-ISCU2 units[18].

While $Fe^{2+}$ binding to ISCU2 is essential for the persulfidation, reduction and cluster synthesis steps, the specific ISCU2 assembly site residues required for Fe coordination have been defined only for chemically reconstituted mouse Fe-ISCU (corresponding to human residues Cys69[ISCU2], Asp71[ISCU2], Cys95[ISCU2], and His137[ISCU2])[19,30] (Table 1). Fe-loaded ISCU2 can bind to $(NIA)_2$ (Fig. 1a), yet the structure of this complex is currently unknown. In structures of related Zn-loaded complexes, metal coordination by Cys69[ISCU2] was found to be replaced

by Cys381[NFS1] in human (Zn-NIAU)$_2$[15] and by Cys138[ISCU2] in (Zn-NIAUX)$_2$[17] (Table 1). Despite the fact that both structures do not represent physiological reaction intermediates, their apparent differences in metal coordination may suggest a structural plasticity of the ISCU2 assembly site. Clearly, this view needs to be verified by elucidation of the functionally relevant Fe coordination during the various biosynthetic stages of persulfide transfer and reduction within the active core ISC complex. Moreover, it is unclear how Fe and persulfide are handled simultaneously by the assembly site residues to facilitate sulfur transfer.

Here, we use a combination of structural, biochemical and spectroscopic approaches to address the molecular mechanism of sulfur transfer during mitochondrial de novo [2Fe-2S] cluster synthesis. We sought to (1) structurally and biochemically identify persulfide intermediates before and after sulfur transfer, (2) understand the molecular role of FXN, (3) identify the ISCU2 assembly site residues involved in sulfur transfer, and (4) define the dynamic changes in Fe coordination during the various reactions. Our current work provides a molecular understanding of both the mechanism and structural dynamics of sulfur transfer within the core ISC complex. Further, it clarifies the molecular role of FXN in this process, which may be relevant to the development of treatments for Friedreich's ataxia.

## Results

### Cryo-EM structures of the core ISC complex reveal heterogeneity at the FXN binding site

To gain structural insights into the persulfide transfer process from Cys381[NFS1] to ISCU2, we prepared a (Fe-NIAUX)$_2$ complex that is able to perform the persulfidation reaction, but is unable to generate the [2Fe-2S] cluster because FDX2 was omitted[14,18,30] (Fig. 1a). To this end, the purified (NIAU)$_2$ subcomplex was mixed under anaerobic conditions with Fe$^{2+}$ and FXN (stoichiometry 1:5:1). Cys was added to initiate persulfidation, and the complex was rapidly vitrified under anaerobic conditions for cryo-EM single particle analysis. A structure determined from this sample reached a global resolution of 2.4 Å for the consensus refinement (C2 symmetry applied; Table 2 and Supplementary Figs. 1–4), and showed a typical (NIAUX)$_2$ complex with an extensive dimeric interface in which the two NFS1 subunits and FXN interacted with ISCU2 via conserved residues[17] (Fig. 1b). In our structure, the density for FXN was weaker in comparison to the (NIAU)$_2$ components, indicating only partial FXN occupancy. Symmetry expansion and subsequent local 3D classification allowed separation of FXN-bound (77%) (Fe-NIAUX)$_2$ and FXN-lacking (23%) (Fe-NIAU)$_2$ states, which were locally refined to resolutions of 2.4 Å and 2.7 Å, respectively (Supplementary Figs. 1 and 2). The partial occupancy of the FXN binding site is best explained by the relatively weak affinity of FXN for the core ISC complex[16,30,33]. This fact was also evident from size exclusion chromatography (SEC) of the (Fe-NIAUX)$_2$ complex. FXN showed partial dissociation from the complex, leading to sub-stoichiometric FXN occupancy (Supplementary Fig. 5).

### Structure of the Fe-coordinated core ISC complex in the absence of FXN

The overall architecture of the (Fe-NIAU)$_2$ complex was similar to that of the published Zn-containing complex[15], yet showed important differences with potential functional implications (Table 2). The ISCU2-bound Fe ion was coordinated by residues Cys381[NFS1], Asp71[ISCU2], Cys95[ISCU2], and likely a water (W1) molecule (Fig. 1c and Table 1). Although persulfidation of NFS1 in the absence of FXN occurred quickly (see below), we did not observe density for a Cys381[NFS1]-bound persulfide at the ISCU2 assembly site. Apparently, the presence of a metal (Fe or Zn) positions the active Cys381[NFS1] residue at the ISCU2 assembly site, whereas without a bound metal the Cys loop of NFS1 is located close to the PLP site[15,18]. Active-site residue His137[ISCU2], although oriented toward the ISCU2 assembly site, is too distant to coordinate

the metal (His-N$^\varepsilon$–Fe distance of 4.8 Å), distinguishing it from the X-ray structure of (Zn-NIAU)$_2$ (Supplementary Fig. 6 and Table 1). Similarly, Cys69[ISCU2] and Cys138[ISCU2] are too far for Fe coordination. The fact that these three residues are not participating in Fe ligation may well explain why the densities for Cys69[ISCU2] and His137[ISCU2] show lower local resolution, indicating higher flexibility.

The Fe-containing conformation raised the important functional question of which Cys residue of ISCU2 may serve as the sulfur acceptor from the Cys loop Cys381[NFS1]. The latter is closest to the sulfur atoms of Cys95[ISCU2] (3.8 Å) and Cys69[ISCU2] (4.3 Å), yet rather distant

**Table 2 | Cryo-EM data collection, processing, model building and validation statistics**

| | (Fe-NIAUX)$_2$ consensus (EMD-17731) | (Fe-NIAU$^S$X)$_2$ (EMD-17732) (PDB 8PK8) | (Fe-N$^S$IAU$^S$X)$_2$ (EMD-17733) (PDB 8PK9) | (Fe-NIAU)$_2$ (EMD-17734) (PDB 8PKA) |
|---|---|---|---|---|
| **Data collection and processing** | | | | |
| Magnification | ×105,000 (nominal) | ×105,000 (nominal) | ×105,000 (nominal) | ×105,000 (nominal) |
| Voltage (kV) | 300 | 300 | 300 | 300 |
| Electron exposure (e$^-$/Å$^2$) | 62 | 62 | 62 | 62 |
| Defocus range (μm) | −0.8 to −2.5 | −0.8 to −2.5 | −0.8 to −2.5 | −0.8 to −2.5 |
| Pixel size (Å) | 0.837 | 0.837 | 0.837 | 0.837 |
| Symmetry imposed | C2 | C1 | C1 | C1 |
| Initial particle images (no.) | 2,189,680 | 2,189,680 | 2,189,680 | 2,189,680 |
| Final particle images (no.) | 696,640 | 507,565 (symmetry expanded) | 393,707 (symmetry expanded) | 269,101 (symmetry expanded) |
| Map resolution (Å) FSC threshold | 2.41 0.143 | 2.49 0.143 | 2.58 0.143 | 2.75 0.143 |
| Map resolution range (Å) | 2.29–3.29 | 2.37–3.43 | 2.41–3.54 | 2.58–3.68 |
| **Refinement** | | | | |
| Initial model used (PDB code) | | 6NZU | 6NZU | 6NZU |
| Model resolution (Å) FSC threshold | | 2.0/2.4 0.143/0.5 | 2.1/2.5 0.143/0.5 | 2.2/2.6 0.143/0.5 |
| Map sharpening B factor (Å$^2$) | −73.3 | a | a | a |
| Model composition | | | | |
| Non-hydrogen atoms | | 6461 | 6409 | 5435 |
| Protein residues | | 800 | 800 | 680 |
| Ligands | | CSS, FE2, PLP, 8Q1 | CSS, FE2, PLP, 8Q1 | FE2, PLP, 8Q1 |
| B factors (Å$^2$) min/max/mean | | | | |
| Protein | | 6.94/135.86/ 37.04 | 2.12/87.13/ 28.96 | 8.66/112.18/ 35.47 |
| Ligand | | 18.96/ 95.18/21.19 | 18.53/ 75.25/28.50 | 11.28/ 87.86/25.04 |
| R.m.s. deviations | | | | |
| Bond lengths (Å) | | 0.007 | 0.004 | 0.006 |
| Bond angles (°) | | 1.322 | 0.668 | 0.861 |
| Validation | | | | |
| MolProbity score | | 1.68 | 1.62 | 1.75 |
| Clashscore | | 6.17 | 6.57 | 6.58 |
| Poor rotamers (%) | | 1.61 | 2.20 | 2.75 |
| Ramachandran plot | | | | |
| Favored (%) | | 96.95 | 98.34 | 97.77 |
| Allowed (%) | | 2.92 | 1.66 | 2.08 |
| Disallowed (%) | | 0.13 | 0.00 | 0.15 |

$^a$Half-maps were subjected to density modification and local anisotropic sharpening using the Phenix software suite (Liebschner et al.[48]).

from Cys138$^{ISCU2}$ (6.5 Å; Table 1, bottom), which might suggest that Cys95$^{ISCU2}$ or Cys69$^{ISCU2}$ would be the most likely candidates for persulfide transfer from Cys381$^{NFS1}$. However, previous biochemical experiments performed under aerobic conditions have suggested Cys69$^{ISCU2}$ and Cys138$^{ISCU2}$ as potential persulfide acceptors[25]. Intriguingly, His137$^{ISCU2}$, which is hydrogen-bonded to the water ligand of Fe between Cys381$^{NFS1}$ and Cys138$^{ISCU2}$, appears to prevent a closer approach of the two Cys residues and to hinder direct sulfur transfer (Supplementary Fig. 6). Therefore, either a substantial conformational change is needed to make Cys138$^{ISCU2}$ available as a direct sulfur acceptor, or one of the other Cys residues may serve as the intermediate or even final sulfur acceptor. Clearly, additional structural and biochemical data under anaerobic conditions are required to resolve this mechanistic problem.

## Structural snapshots of pre- and post-sulfur-transfer states in (Fe-NIAUX)$_2$ complexes

For the 3D class showing clear density for FXN, 3D refinement gave rise to a map with density for two alternate conformations of the catalytic Cys loop of NFS1, i.e., an "inward" conformation in which Cys381$^{NFS1}$ was oriented toward the PLP cofactor of NFS1 and an "outward" conformation in which Cys381$^{NFS1}$ faced the ISCU2 assembly site. This arrangement differed from the (Zn-NIAUX)$_2$ cryo-EM structure[17], where the Cys loop of NFS1 was positioned in an "intermediate" conformation. Using a multi-reference 3D classification approach (see "Methods"), followed by local refinements, our alternate conformations were separated and resolved at 2.5 Å (inward) and 2.6 Å (outward) (Fig. 1d, e, Table 2 and Supplementary Fig. 1). Most importantly, the inward conformation, henceforth referred to as (Fe-NIAU$^S$X)$_2$, contained additional density at Cys138$^{ISCU2}$ that could be confidently modeled as a persulfide (Fig. 1d, right). This likely represents the post-sulfur-transfer state, in which the persulfide sulfur of Cys138$^{ISCU2}$ (CSS138) ligates the Fe center together with Asp71$^{ISCU2}$, Cys95$^{ISCU2}$ and likely a water (W1) molecule (Table 1). How Cys138$^{ISCU2}$ may have become available to accept the sulfur from Cys381$^{NFS1}$, despite their large distance in the (Fe-NIAU)$_2$ structure (see above), will be presented below. Apparently, after sulfur transfer, the Cys loop of NFS1 has moved away from the ISCU2 assembly site, thereby positioning Cys381$^{NFS1}$ again close to the PLP site.

In the outward state (hereafter termed (Fe-N$^S$IAU$^S$X)$_2$), we also observed additional densities that could be modeled as persulfides at both Cys381$^{NFS1}$ (CSS381) and Cys138$^{ISCU2}$ (CSS138) (Fig. 1e, right). Both persulfides are within Fe coordination distance and thus bind the metal together with Asp71$^{ISCU2}$ and Cys95$^{ISCU2}$ (Table 1). The presence of two persulfides was unforeseen. This arrangement could either represent both pre- and post-sulfur-transfer states (due to a second round of persulfidation), or be explained by a mixture of particles in either pre- or post-sulfur transfer states. Generally, the detection of a pre-sulfur-transfer state was unexpected due to its potentially short-lived character, and may indicate an incomplete sulfur transfer reaction or an equilibrium between the two states as suggested by findings below. Collectively, these structural snapshots of intermediate states before and after sulfur transfer from NFS1 to ISCU2 provide a view on the possible mechanism of sulfur trafficking (see below), yet do not readily explain why Cys138$^{ISCU2}$ is the preferred persulfide acceptor.

## FXN binding rearranges the ISCU2 assembly site to facilitate persulfide transfer

Comparing our (Fe-NIAU)$_2$ and the two (Fe-NIAUX)$_2$ structures allowed insights into the dedicated mechanistic role of FXN. Upon FXN binding, the overall architecture of the (Fe-NIAU)$_2$ complex remained virtually unchanged, with the exception of significant local conformational alterations at the ISCU2 assembly site. Both Cys69$^{ISCU2}$ and His137$^{ISCU2}$ are shifted substantially away from the ISCU2-bound Fe, as previously observed in the (Zn-NIAUX)$_2$ complex[17], and now form contacts with FXN in both the outward- and inward-facing loop

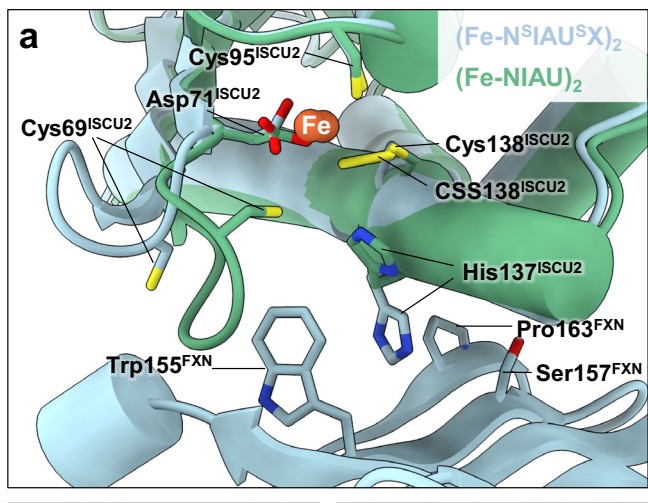

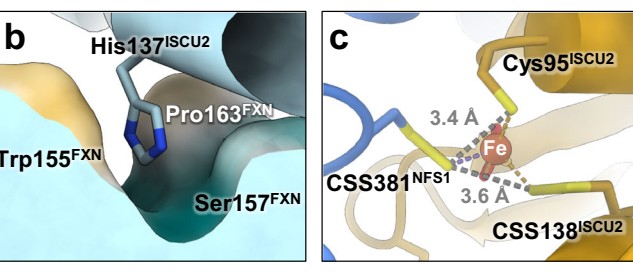

**Fig. 2 | Structural rearrangements of the ISCU2 assembly site upon FXN binding facilitate persulfide transfer. a** Overlay of the (Fe-NIAU)$_2$ and (Fe-N$^S$IAU$^S$X)$_2$ structures shows the structural changes upon FXN binding (Cys381$^{NFS1}$ not shown for figure clarity). While the constantly Fe$^{2+}$-coordinating Cys95$^{ISCU2}$ and Asp71$^{ISCU2}$ hardly change positions, Cys138$^{ISCU2}$ moves toward the Fe$^{2+}$ and now coordinates the metal. Concomitantly, His137$^{ISCU2}$ and Cys69$^{ISCU2}$ are shifted away from the ISCU2 assembly site, and His137$^{ISCU2}$ is now accommodated in a pocket formed by residues including Trp155$^{FXN}$, Ser157$^{FDX}$ and Pro163$^{FDX}$ (**b**). **c** The pre-sulfur transfer state of the (Fe-N$^S$IAU$^S$X)$_2$ structure could conceivably allow for sulfur delivery from Cys381$^{NFS1}$ to both Cys95$^{ISCU2}$ (3.4 Å distance) and Cys138$^{ISCU2}$ (3.6 Å distance).

structures (Fig. 2a and Table 1). This makes sulfur acceptance by Cys69$^{ISCU2}$ unlikely. Particularly, His137$^{ISCU2}$, which appears to block close contact of Cys381$^{NFS1}$ and Cys138$^{ISCU2}$ in our (Fe-NIAU)$_2$ structure, becomes inserted into a pocket formed by FXN residues Trp155, Ser157, and Pro163, thereby removing the steric separation of Cys381$^{NFS1}$ and Cys138$^{ISCU2}$ (Fig. 2b and Supplementary Movie 1). As a consequence, in the (Fe-N$^S$IAU$^S$X)$_2$ structure, CSS138 is located close to both the Fe (Fe-S distance 2.3 Å) and CSS381 (S-S distance 3.6 Å) (Table 1). Even in this arrangement the distance between the pre-sulfur-transfer persulfide CSS381 and the Cys95$^{ISCU2}$ sulfur (3.4 Å) is slightly shorter than to CSS138, raising the question of whether Cys95$^{ISCU2}$ may serve as an intermediate sulfur-accepting moiety before sulfur transfer to Cys138$^{ISCU2}$. In any case, the structural changes triggered by FXN binding move persulfide CSS381 into a position allowing sulfur transfer to ISCU2. The relatively open coordination sphere at the assembly site of the (Fe-N$^S$IAU$^S$X)$_2$ complex may provide enough space for the sulfur transfer reaction by nucleophilic attack by either Cys95$^{ISCU2}$ or Cys138$^{ISCU2}$ sulfurs (Fig. 2c). The observed effects of FXN binding in moving the sulfur-donating and -accepting Cys residues into optimal positions for possible sulfur transfer may explain previous biochemical observations that FXN substantially accelerates the rate of persulfidation within the (NIAU)$_2$ complex[25,26].

## Only ISCU2 assembly site residues Asp71, Cys95, and Cys138 are essential for sulfur transfer

To answer the question of whether persulfide transfer to Cys138$^{ISCU2}$ in the (Fe-NIAU$^S$X)$_2$ complex occurs directly or indirectly via Cys95$^{ISCU2}$,

we first analyzed the relative importance of the five ISCU2 assembly site residues for the sulfur transfer reaction by employing a strictly anaerobic Cys alkylation-based gel-shift assay (see reaction scheme in Supplementary Fig. 7a)[18]. Persulfidation of wild-type (WT) and various assembly-site ISCU2 mutant proteins was initiated by Cys and FXN addition to (Fe-NIAU)$_2$, and the reaction was quenched at different time points by consecutive addition of maleimide-polyethylene glycol$_{11}$-biotin (MPB, for labeling of native and persulfidated Cys residues) and the denaturing agent SDS. Reductive cleavage of disulfides by TCEP reducing agent decreased the mass of persulfidated ISCU2 as detected by SDS-PAGE. Quantitation showed that already after 10 s, 75% of maximal mono-persulfidation (1x SSH; maximum of 86% reached after 2 min) had occurred (Supplementary Fig. 7b, c). At a much slower rate, a steady increase of doubly persulfidated (2x SSH) ISCU2 was seen, which likely represented a non-physiological side reaction. To avoid this unspecific persulfidation, further reactions were incubated for only 10 s.

ISCU2 variants C69S, H137A (both assembly-site residues) and C130S (a non-conserved Cys residue) yielded 70–101% persulfide formation compared to ISCU2 WT (Fig. 3a and Supplementary Fig. 8a–c), clearly showing that these residues are not essential for persulfidation. Since Fe was essential for this reaction (Fig. 3a), Cys69$^{ISCU2}$ and His137$^{ISCU2}$ appeared not to serve as Fe ligands during sulfur transfer, despite the fact that they are involved in the initial Fe binding to ISCU2 (Supplementary Fig. 9)[19]. This finding is consistent with our (Fe-NIAU)$_2$ structure, in which neither Cys69$^{ISCU2}$ nor His137$^{ISCU2}$ coordinate the Fe. A previous study reported that the mouse ISCU-C69S variant abrogates persulfidation, and a function of this residue in sulfur transfer was suggested[25]. Based on our structural data the effect is more likely explained by the Fe coordination function of this ISCU2 residue (Table 1). For free ISCU2, Fe (or Zn) binding strictly requires Cys69 as a ligand (Supplementary Fig. 9)[30], as opposed to NFS1-bound ISCU2 (Fig. 1c–e). Because the previous study had performed ISCU-C69S persulfidation with as-purified, non-metal reconstituted protein[25], distorted metal binding may have disabled persulfidation.

In contrast to the ISCU2 variants described above, the ISCU2-D71A, -C95S, and -C138S assembly-site variants and Cys double mutant proteins produced only background levels of persulfide (Fig. 3a and Supplementary Fig. 8b, d, e). Since residues Asp71$^{ISCU2}$ and Cys95$^{ISCU2}$ serve as metal ligands in all metal-containing (NIAU)$_2$ and (NIAUX)$_2$ structures including those presented here (Table 1), the sulfur transfer defect of these variants can satisfactorily be explained by a lack of Fe coordination during persulfidation. The failure of the ISCU2-C138S variant could be a dual effect, first by lack of metal binding, and second by its proposed role as a persulfide-receiving residue. Collectively, only three of the five essential ISCU2 assembly site residues are required for the sulfur transfer step.

## Rapid, specific and direct persulfidation of ISCU2 Cys138 by NFS1

Previous mass spectrometric (MS) analyses showing that Cys69$^{ISCU2}$ and Cys138$^{ISCU2}$ may act as persulfide acceptors[25] were carried out under aerobic conditions using as-purified, Zn-bound mouse ISCU and reaction times on the order of 3 min (which favors unspecific persulfidation; cf. Supplementary Fig. 7c). Here, we developed a rapid MS-based persulfide detection approach using the Fe-loaded (Fe-NIAUX)$_2$ complex under anaerobic conditions. Isobaric thiol labeling reagents of different masses allowed quantitation of the persulfide levels on each individual ISCU2 Cys residue on a time scale of seconds. Thereby, the assay can also determine transient persulfidation (see scheme in Supplementary Fig. 10a). Briefly, Cys-dependent persulfide formation on (Fe-NIAUX)$_2$ was quenched with a mixture of denaturing SDS detergent, the metal chelator EDTA, and the iodoacetyl tandem mass tag (iodoTMT) labeling reagent. As a control, CD spectroscopy was used to verify that Fe dissociated rapidly and quantitatively from Fe-loaded ISCU2, thereby documenting the efficiency of terminating

persulfidation (Supplementary Fig. 10b, c). Excess label was quenched by Cys addition, and after reductive cleavage of persulfides, a second iodoTMT reagent with different mass was used to label the cleavage products (Supplementary Fig. 10a). Persulfide adducts at each Cys-containing peptide were quantified by Nano HPLC-MS$^2$ analysis of trypsin-digested samples. We first analyzed persulfidation of NFS1, which occurred at a high rate and showed specific labeling of only the active-site Cys381$^{NFS1}$ independently of Fe$^{2+}$ (Supplementary Fig. 11a, b).

Similarly, ISCU2 was persulfidated with high rate and specificity at Cys138$^{ISCU2}$, yet unlike NFS1 this happened in a completely Fe-dependent manner (Fig. 3b, c). In the absence of Fe, all Cys residues were labeled only after longer incubation times and rather inefficiently, clearly indicating a non-physiological effect. The rapid persulfidation of Cys138$^{ISCU2}$ was in good agreement with our MPB-based persulfidation assay (cf. Supplementary Fig. 7b, c). The lower amount of Cys138$^{ISCU2}$ persulfide that was formed in the iodoTMT assay (ca. 50%) compared to the MPB assay (up to 90%) likely corresponds to the lower stoichiometry of (NIA)$_2$ and FXN over ISCU2 (1:1:1, as opposed to 2:2:1 in the MPB assay), which was chosen to minimize the amount of costly iodoTMT reagent. Interestingly, specific persulfidation of Cys138$^{ISCU2}$ was also observed in samples lacking FXN, yet at much slower rates, while NFS1 was modified at a normal high rate (Fig. 3d and Supplementary Fig. 11c). This observation fits well to the slow FXN-independent [2Fe-2S] cluster formation in vitro (detected by the CD-based Fe/S cluster biosynthesis assay[14,18]) indicating that FXN is not essential for [2Fe-2S] cluster assembly, yet substantially increases its rate. Moreover, the non-essential character of FXN is evident in *S. cerevisiae*, where *YFH1* is the only gene of the core ISC system that can be deleted without losing cell viability[34,35].

No sulfur transfer intermediate on Cys95$^{ISCU2}$ was observed, despite the fact that in the (Fe-N$^S$IAU$^S$X)$_2$ cryo-EM structure this residue was as close to the sulfur-donating Cys381$^{NFS1}$ as Cys138$^{ISCU2}$ (Figs. 1e and 2c and Table 1). To accumulate a potential Cys95$^{ISCU2}$ intermediate, we used the ISCU2-C138S variant and performed the persulfidation reaction. No labeling of Cys95$^{ISCU2}$ (or other ISCU2 Cys residues) above background was detectable (Fig. 3a and Supplementary Fig. 11e). As an internal control, Cys381$^{NFS1}$ was persulfidated as in wild-type controls. We finally examined whether the potential Cys95$^{ISCU2}$ intermediate was converted too rapidly for the Cys138$^{ISCU2}$ persulfide to be detectable by our experimental setup. We therefore performed the iodoTMT assay at 0 °C to slow down the reaction, and we measured the first data point already after <1 s of persulfidation ((NIA)$_2$:FXN:ISCU2 = 2:2:1). Under these conditions, persulfidation of Cys138$^{ISCU2}$ was much slower and reached the levels obtained at room temperature only after 5 min, while Cys381$^{NFS1}$ persulfidation was still virtually complete after <1 s (Fig. 3e and Supplementary Fig. 11d). Clearly, no other ISCU2 Cys residues were modified above background, making persulfidation intermediates including that of Cys95$^{ISCU2}$ unlikely. We conclude that Cys138$^{ISCU2}$ is the rapid, specific, and direct sulfur acceptor from Cys381$^{NFS1}$ upon substantial FXN-facilitated structural changes of the ISCU2 assembly site to properly position the sulfur-donating and -accepting residues around the Fe ion.

## Mössbauer spectroscopy identifies a FXN- and persulfidation-shifted equilibrium of two ISCU2-bound Fe$^{2+}$ species

The findings above provide evidence for a highly dynamic ISCU2 assembly site that undergoes multiple conformational changes during Cys381$^{NFS1}$ persulfide binding to ISCU2-bound Fe, FXN-mediated (Fe-NIAU)$_2$ rearrangement, and subsequent sulfur transfer to Cys138$^{ISCU2}$. We used Mössbauer spectroscopy to further elucidate the dynamic alterations in Fe coordination at the ISCU2 assembly site. With the exception of Fe binding to free murine ISCU[19], the ISCU2 Fe coordination and its dynamics during the intermediate steps of de novo [2Fe-2S] cluster formation have not been analyzed. We recorded Mössbauer spectra for different biochemically generated reaction intermediates,

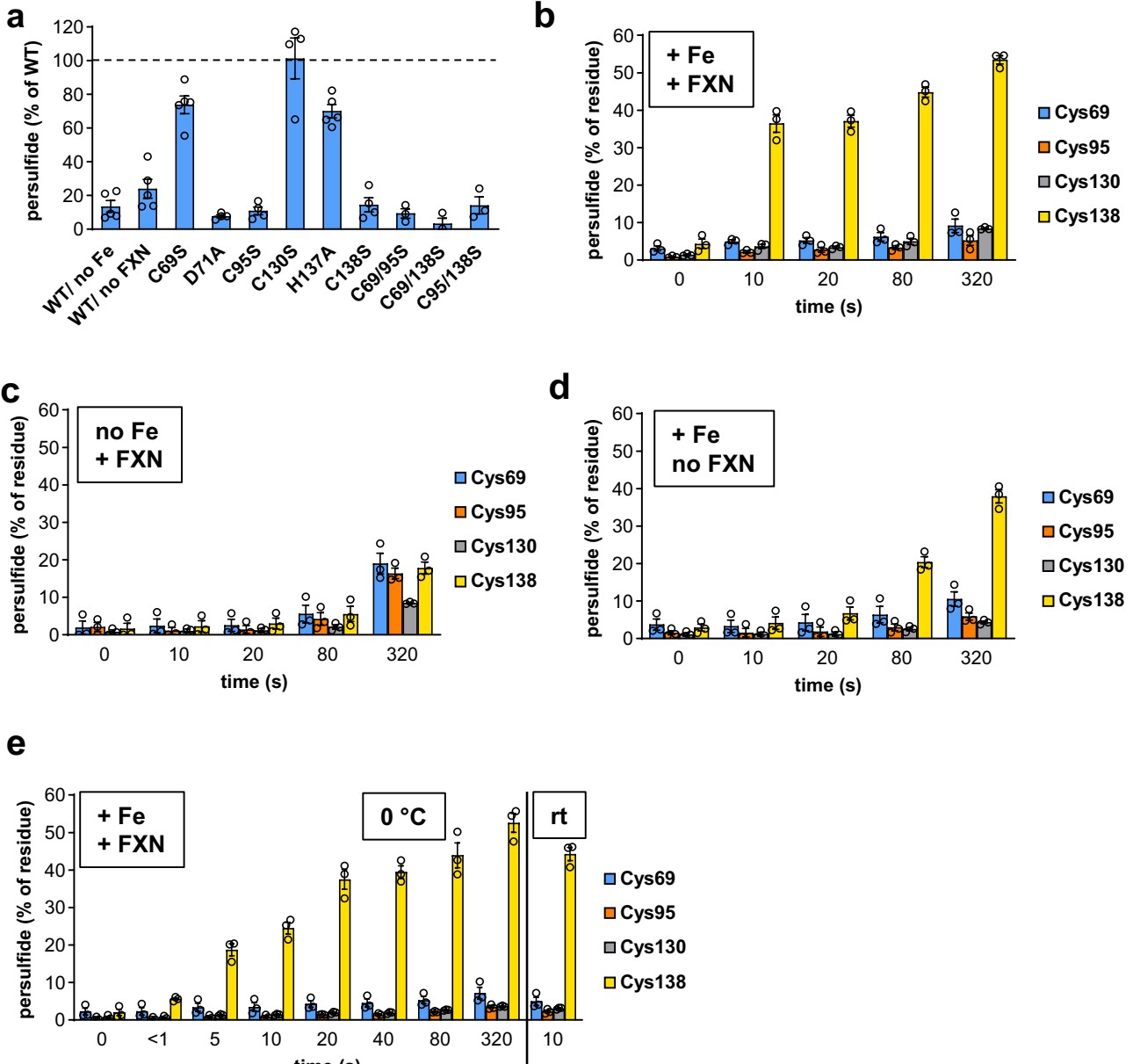

**Fig. 3 | ISCU2 residues Asp71, Cys95, and Cys138 are required for the metal-dependent, rapid and specific persulfidation of Cys138. a** In vitro persulfidation of ISCU2 by the cysteine desulfurase NFS1 was investigated via a MPB alkylation-based gel-shift assay (Supplementary Fig. 7). In reactions containing 20 μM ISCU2, 20 μM (NIA)$_2$, 40 μM FXN, as well as excess Fe$^{2+}$ and ascorbate, persulfide formation was initiated by addition of excess Cys. Reactions performed at room temperature (rt) were quenched after 10 s by addition of MPB and SDS. Persulfidation of ISCU2 mutant proteins and control reactions with wild-type (WT) ISCU2 lacking Fe or FXN was quantified by densitometry of the corresponding bands in SDS-PAGE gels (Supplementary Fig. 8), and normalized to control reactions with WT ISCU2 (dotted line). "WT/ no Fe" reactions contained no Fe$^{2+}$ and ascorbate but 1 mM DTPA to chelate Fe. Error bars indicate the SEM (for WT/ no Fe, WT/ no FXN, C69S, H137A: $n = 5$; for D71A, C95S, C130S, C138S: $n = 4$; for others: $n = 3$; independent

experiments). **b–e** Persulfidation of individual Cys residues of WT ISCU2 was investigated via an iodoTMT-based isobaric alkylation approach (Supplementary Fig. 10). Reactions containing 20 μM ISCU2, 20–40 μM of (NIA)$_2$ and optionally FXN, and either 100 μM Fe$^{2+}$ (**b, d, e**) or 500 μM DTPA (no Fe) (**c**) were quenched with excess iodoTMT (from stocks containing SDS and EDTA) at the indicated time points after addition of 100 μM Cys. Reactions were performed at room temperature (**b–d**) or at 0 °C (**e**). Following TCEP-mediated persulfide cleavage, samples were labeled with a second iodoTMT reagent. Trypsin-digested samples were analyzed by nanoHPLC-MS$^2$, and iodoTMT reporter fragment signals of both labels were integrated to quantitate the relative amount of persulfidation of individual Cys residues. Error bars indicate the SEM ($n = 3$; independent experiments). Source data are provided as a Source Data file.

in particular the $^{57}$Fe-loaded complexes (Fe-NIAU)$_2$, (Fe-NIAUX)$_2$ (no Cys added), and persulfidated (Fe-NIAU$^S$X)$_2$ (Fig. 4 and Table 3). As controls we used $^{57}$Fe-loaded ISCU2 (Fe-U)[19] (Fig. 4a), (NIAX)$_2$ plus added $^{57}$Fe (lacking ISCU2; Fig. 4e), and the final product, i.e., mature [2Fe-2S] cluster-loaded ISCU2 (Fig. 4f).

Various $^{57}$Fe species were identified by Mössbauer spectroscopy across the different samples, and were grouped according to their

isomer shifts $\delta$. Two distinct Fe$^{2+}$ species (components 1 and 2) were not present in the $^{57}$Fe + (NIAX)$_2$ sample, hence being specific for ISCU2 (Fig. 4a–e and Table 3). Conversely, the Fe$^{3+}$/Fe$^{2+}$ species observed in (NIAX)$_2$ (represented by component A with $\delta = 0.30$ mms$^{-1}$ and component B with $\delta = 1.24$ mms$^{-1}$) were not related to ISCU2. Component A (but not B) was present in the various (NIAU)$_2$ complexes, yet could be minimized by optimizing the ionic conditions for assembly of these

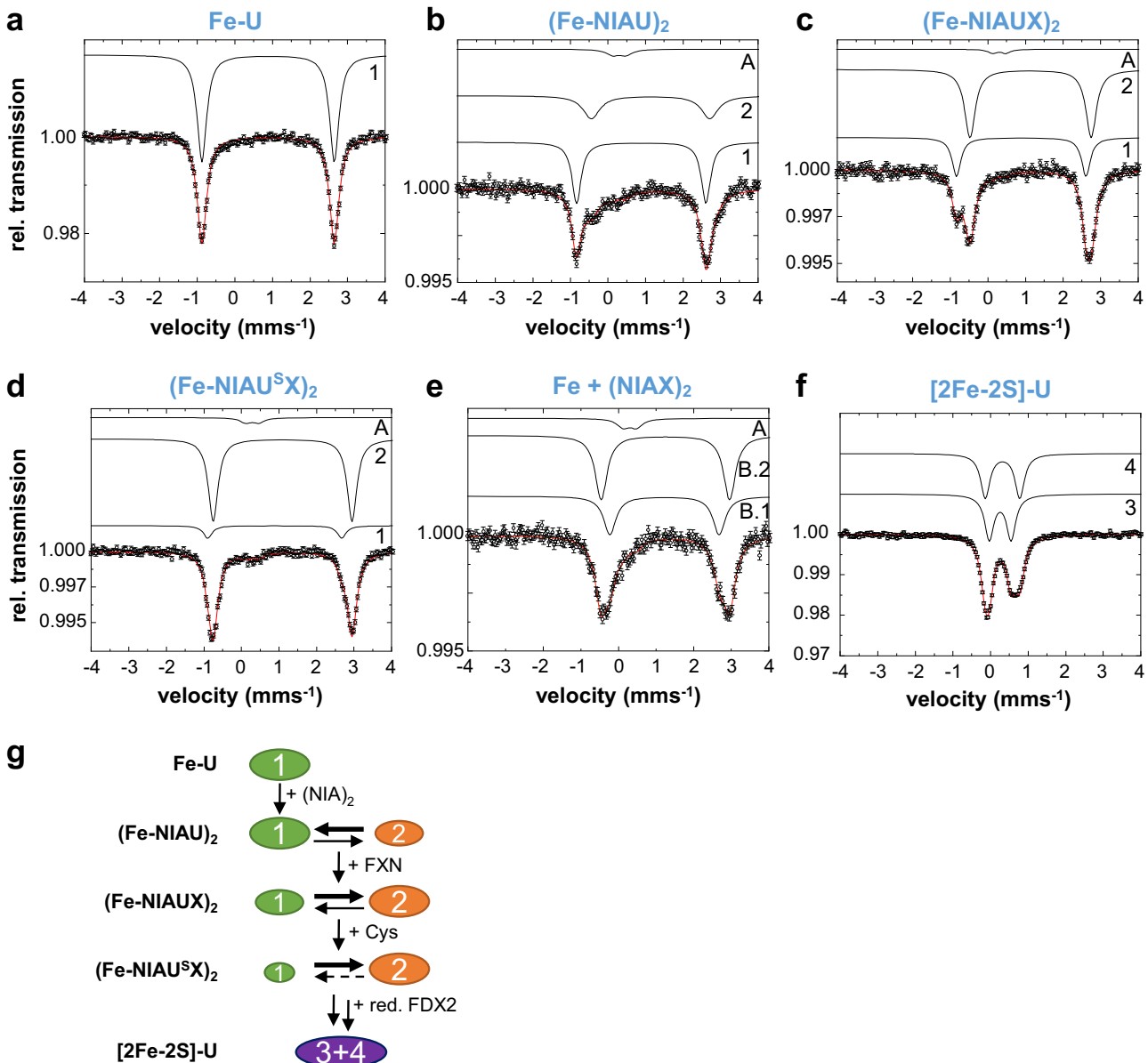

**Fig. 4 | Mössbauer spectroscopy reveals an equilibrium of two distinct Fe coordination states in intermediates of [2Fe-2S] cluster synthesis on ISCU2.** Spectra of the indicated complexes reconstituted with 1 eq. $(NH_4)_2{}^{57}Fe(SO_4)_2$ per ISCU2. For the detailed sample composition, see Supplementary Table 2a. U: ISCU2; X: FXN; F: FDX2. **a** Mössbauer spectrum recorded for $^{57}Fe$-loaded ISCU2 (Fe-U). **b–d** Mössbauer spectra of samples containing (NIA)_2 and/or FXN in excess over ISCU2. Cysteine was added as the last component in (**d**) to initiate ISCU2 persulfidation. **e** Mössbauer spectrum in the presence of (NIAX)_2, lacking ISCU2. **f** Mössbauer spectrum of enzymatically reconstituted [2Fe-2S]-ISCU2 from a

reaction containing (NIA)_2, FXN, FDX2 and FDXR in catalytic amounts, as well as Cys and NADPH. Each Mössbauer spectrum was recorded at 77 K and zero applied external field. The red lines display the best fit of the data using the components shown as black lines with parameters given in Table 3. Mössbauer data points are presented as relative transmission per velocity channel derived from detector counts (see "Methods"). Error bars indicate ±SD. **g** Scheme depicting the shift in equilibrium of ISCU2-bound Fe-species (components 1 and 2) by FXN binding and persulfidation (+Cys) during consecutive stages of [2Fe-2S] cluster biosynthesis on ISCU2 (Fe components 3 and 4) identified by Mössbauer spectroscopy.

complexes (compare Fig. 4 and Table 3 with Supplementary Fig. 12 and Supplementary Table 1; conditions in Supplementary Table 2a, b). As a further control, we analyzed ISCU2 after enzymatic [2Fe-2S] cluster reconstitution with catalytic amounts of (NIA)_2, FXN, and the electron transfer chain NADPH-FDX2-FDXR[14]. Two distinct $^{57}Fe/S$ cluster-related components (3 and 4, Fig. 4f and Table 3) were detected with parameters comparable to those published for mouse [2Fe-2S]-ISCU[30], yet were clearly distinct from the ISCU2-related components 1 and 2, showing the specificity of the latter Fe species for (NIAU)_2 samples.

The ISCU2-specific component 1 (isomer shift $\delta = 0.89$, quadrupole splitting $\Delta E_Q = 3.4$–$3.6$ mms$^{-1}$; Table 3) was the only species present in Fe-U and indicated high-spin $Fe^{2+}$ ligated by 1–2 S and 2–

3 N/O ligands. This finding is consistent with the established tetrahedral metal coordination by Cys69$^{ISCU2}$, Asp71$^{ISCU2}$, Cys95$^{ISCU2}$, and His137$^{ISCU2}$ in the NMR structure of Zn-bound murine ISCU (PDB 1WFZ) (Fig. 4a and Supplementary Fig. 9)[19,30]. Strikingly, the amount of component 1 in the various (NIAU)_2 complexes successively decreased along the Fe/S biosynthetic pathway. Concomitantly, component 2 (isomer shift $\delta = 1.10$–$1.14$, quadrupole splitting $\Delta E_Q = 3.15$–$3.70$ mms$^{-1}$; Table 3) amounts increased, particularly upon (NIA)_2 binding to Fe-U from 0 to 38%, and further upon FXN addition ((Fe-NIAUX)_2) from 38 to 66% (Fig. 4a–d, g and Table 3). After persulfidation upon Cys addition ((Fe-NIAU$^S$X)_2 intermediate), component 2 further increased from 66 to 83%. Notably, the isomer shift

**Table 3 | Mössbauer parameters of $^{57}$Fe-bound ISCU2 or various ISC complex intermediates**

| | Component 1 | | | | Component 2 | | | | Component A | | | |
|---|---|---|---|---|---|---|---|---|---|---|---|---|
| | $\delta$ | $\Delta E_Q$ | $\Gamma$ | $A^2$ (%) | $\delta$ | $\Delta E_Q$ | $\Gamma$ | $A^2$ (%) | $\delta$ | $\Delta E_Q$ | $\Gamma$ | $A^2$ (%) |
| Fe-U | 0.89 | 3.52 | 0.30 | 100 | | | | | | | | |
| (Fe-NIAU)$_2$ | 0.89 | 3.45 | 0.30 | 55 | 1.14 | 3.15 | 0.54 | 38 | 0.30 | 0.35 | 0.41 | 7 |
| (Fe-NIAUX)$_2$ | 0.89 | 3.44 | 0.28 | 31 | 1.14 | 3.22 | 0.34 | 66 | 0.30 | 0.35 | 0.28 | 3 |
| (Fe-NIAU$^S$X)$_2$ | 0.89 | 3.57 | 0.28 | 11 | 1.10 | 3.70 | 0.31 | 83 | 0.30 | 0.35 | 0.36 | 6 |
| | Component A | | | | Component B | | | | | | | |
| Fe + (NIAX)$_2$ | 0.30 | 0.35 | 0.40 | 8 | 1.24 | 2.91 | 0.40 | 35 | | | | |
| | | | | | 1.25 | 3.42 | 0.38 | 57 | | | | |
| | Component 3 | | | | Component 4 | | | | | | | |
| [2Fe-2S]-U | 0.26 | 0.58 | 0.29 | 50 | 0.32 | 0.92 | 0.30 | 50 | | | | |

Isomer shift ($\delta$), quadrupole splitting ($\Delta E_Q$) and line width ($\Gamma$) are given in mms$^{-1}$ with an error of ±0.02 mms$^{-1}$. The relative spectral area $A^2$ is given in % and has an error of ±3%.

slightly decreased to $\delta = 1.10$ mms$^{-1}$, and the quadrupole splitting significantly increased to 3.70 mms$^{-1}$. Whether the latter increase is due to a rearrangement of the Fe ligand environment (for instance, dissociation of Cys381$^{NFS1}$ from the Fe and binding of Cys138$^{ISCU2}$) or an effect induced by persulfidation remains unclear. These findings suggested that (NIAU)$_2$ complex formation, FXN binding and persulfidation successively led to a transition toward component 2, thereby indicating substantial changes in Fe coordination at the ISCU2 assembly site during biosynthesis, with (at least) two distinct Fe-coordination modes (Fig. 4g and Table 3).

Component 1 represented the major ISCU2-specific Fe species (55%) in (Fe-NIAU)$_2$ (Table 3). When compared to the Fe coordination in Fe-U, our (Fe-NIAU)$_2$ cryo-EM structure showed Cys69$^{ISCU2}$ being replaced by Cys381$^{NFS1}$, and His137$^{ISCU2}$ being exchanged for a water molecule (Fig. 1c). This substantial rearrangement upon Fe-U binding to (NIA)$_2$ may, however, not be detectable by Mössbauer spectroscopy, because the respective numbers of S- and N/O-ligands remained unchanged. Interestingly, the movement of His137$^{ISCU2}$ away from the Fe coordination site may partially mimic the alterations observed upon FXN binding, and generally shows the high structural dynamics of the ISCU2 assembly site. Its plasticity is also evident from comparison to the crystal structure of (Zn-NIAU)$_2$, where metal ligation via His137$^{ISCU2}$ was maintained, in addition to coordination by Cys381$^{NFS1}$, Asp71$^{ISCU2}$, Cys95$^{ISCU2}$, i.e., similar as in Fe-U but with an exchange of Cys69$^{ISCU2}$ for Cys381$^{NFS1}$ (Table 1 and Supplementary Fig. 6).

Component 2 with $\delta = 1.10–1.14$ mms$^{-1}$ constituted the major fraction of ISCU2-specific Fe-species in both FXN-containing (Fe-NIAUX)$_2$ and (Fe-NIAU$^S$X)$_2$ complexes (66% and 83%, respectively; Fig. 4c, d and Table 3). The tetrahedral (2S-1N-1O) coordinated Fe$^{2+}$ site of Fe-U represented by component 1 shows $\delta = 0.89$ mms$^{-1}$. A possible bidentate binding of Asp71 increases the coordination number by one. This would lead to an increase in the isomer shift by ~0.1 mms$^{-1}$ [36,37] leading to ~1 mms$^{-1}$ for a (2S-1N-2O) coordination. Changing His137$^{ISCU2}$ to a water ligand as observed by our cryo-EM data would not affect the isomer shift much. Thus, Mössbauer spectroscopy indicates a bidentate binding of Asp71 which could not be assigned with certainty from the obtained resolution of the cryo-EM structures (Fig. 1). Moreover, an octahedral (2S-4O) coordination of Fe$^{2+}$ with a further N/O ligand (e.g., water) would lead to a further increase in the isomer shift by ~0.1 mms$^{-1}$ resulting in 1.1 mms$^{-1}$, close to the values experimentally observed for component 2 in the ISC complexes, in particular for (Fe-NIAU$^S$X)$_2$. However, we note that an octahedral coordination was not observed in the (Fe-NIAU$^S$X)$_2$ cryo-EM structure, where bidentate Asp71$^{ISCU2}$, Cys95$^{ISCU2}$, CSS138$^{ISCU2}$, and one water (i.e., 2S-3O) have been assigned as Fe ligands (Fig. 1d). Notably, the line width $\Gamma$ of component 2 in Mössbauer analyses (Table 3) is significantly decreased (by 40%) in (Fe-NIAUX)$_2$ and (Fe-NIAU$^S$X)$_2$ compared to (Fe-NIAU)$_2$ samples, suggesting that FXN binding structurally rigidifies this Fe species.

The presence of a substantial amount (31%) of component 1 identified by our Mössbauer analysis of (Fe-NIAUX)$_2$ may be related to the dynamic association and dissociation of FXN. Such behavior is suggested by our work (Supplementary Fig. 5), NMR data showing only 65% of FXN being bound to (Zn-NIAU)$_2$ under stoichiometric conditions [30], and recent biochemical studies [33]. Hence, in our (Fe-NIAUX)$_2$ Mössbauer sample, the fraction of non-FXN-bound (Fe-NIAU)$_2$ complex likely accounts for the Mössbauer parameters of component 1. Collectively, these findings suggest an equilibrium between (at least) two distinct core ISC complex states with different Fe coordination (Fig. 4g and Table 3). The first state (component 1) predominates in the (Fe-NIAU)$_2$ sample, where Fe is ligated by Cys381$^{NFS1}$, Asp71$^{ISCU2}$, Cys95$^{ISCU2}$ and a water molecule (as in our (Fe-NIAU)$_2$ structure) in a tetrahedral fashion (stage 2 in Fig. 5). In contrast, component 2 prevails upon FXN binding, e.g., in (Fe-NIAU$^S$X)$_2$. The fact that the isomer shift of component 2 by Mössbauer data is not fully consistent with the observed Fe coordination in our (Fe-NIAU$^S$X)$_2$ structure may be due to a malleable ISCU2 assembly site, the presence of additional water molecules not resolved by cryo-EM, and/or the drastically different experimental conditions required for cryo-EM and Mössbauer experiments including pH differences (see "Methods"). At any rate, the equilibrium between the two different Fe binding states can be shifted by FXN binding and persulfidation suggesting physiological relevance.

## Discussion

Fe/S protein biogenesis in eukaryotes starts with the de novo synthesis of [2Fe-2S] clusters inside mitochondria, and involves several steps: (1) the sulfur release from free Cys by the cysteine desulfurase complex NFS1-ISD11-ACP1, (2) FXN-stimulated transfer of the NFS1-bound persulfide to the scaffold protein ISCU2 which has Fe$^{2+}$ bound, (3) the FDX2-dependent reduction of the persulfide to sulfide, and finally, (4) the formation of the [2Fe-2S] cluster by interaction of two [1Fe-1S]-containing ISCU2 on different core ISC complexes [14,15,18,25,26,30]. To date, many of the molecular mechanistic details of how Fe and sulfur are handled by the ISC proteins generating the [2Fe-2S] cluster are poorly understood. We have used a combination of structural biology, biochemistry, and Mössbauer spectroscopy under anaerobic conditions to structurally and biochemically characterize various intermediates of this multi-stage process including those containing the transient persulfides. Thereby, we could define the pathway of the persulfide sulfur from Cys381$^{NFS1}$ to the receiving Cys138$^{ISCU2}$ and elucidate the conformational rearrangements of the five essential ISCU2 active-site residues in this sulfur-trafficking process and in ISCU2-bound Fe coordination.

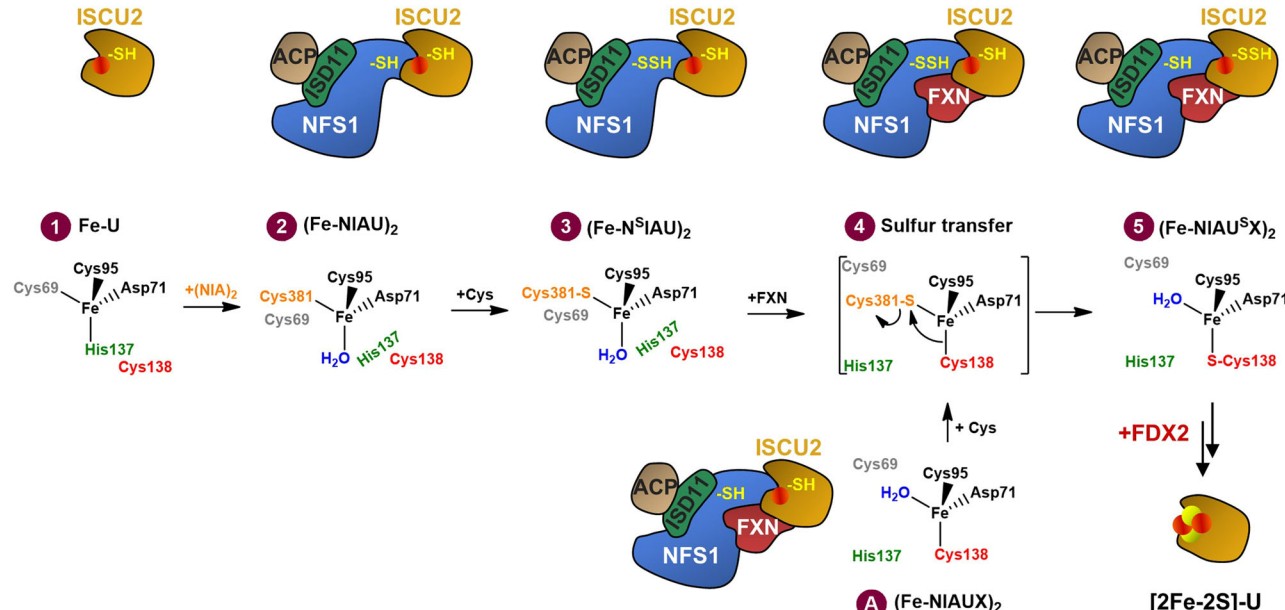

**Fig. 5 | Model of Fe and S binding by ISCU2 at consecutive stages of de novo [2Fe-2S] cluster biosynthesis.** Top: Cartoon of the ISC complexes analyzed in this study. Bottom: Coordination of Fe by the five residues of the ISCU2 assembly site, Cys381$^{NFS1}$ (orange), and water (blue) at different stages of the biosynthetic pathway. (1) Free ISCU2 (U) tetrahedrally coordinates Fe$^{2+}$ via its residues Cys69, Asp71, Cys95 and His137. (2) Binding of Fe-U to (NIA)$_2$ leads to an Fe coordination in which Cys69$^{ISCU2}$ is replaced by Cys381$^{NFS1}$ and His137 by water. (3) Cys addition creates the Cys381$^{NFS1}$ persulfide which also can coordinate the Fe. In this state, steric hindrance by His137$^{ISCU2}$ and the distal location of Cys138$^{ISCU2}$ preclude persulfide transfer from Cys381$^{NFS1}$ to Cys138$^{ISCU2}$. (4) Dynamic binding of FXN to (Fe-N$^S$IAU)$_2$ promotes His137$^{ISCU2}$ to move away from the assembly site, while Cys138$^{ISCU2}$ moves closer to the Fe and Cys381-S$^{NFS1}$. (5) Thereby, Cys138$^{ISCU2}$ coordinates the Fe as a prerequisite for direct persulfide acceptance from Cys381$^{NFS1}$ to form the (Fe-NIAU$^S$X)$_2$ intermediate. Despite the fact that Cys95$^{ISCU2}$ is within a similar distance from the Cys381$^{NFS1}$ persulfide as Cys138$^{ISCU2}$, no persulfidation is observed. (A) As a rather unlikely alternative (see Discussion), the preformed (Fe-NIAUX)$_2$ complex (similar to the (Zn-NIAUX)$_2$ structure[17]) may coordinate the Fe by Asp71$^{ISCU2}$, Cys95$^{ISCU2}$, Cys138$^{ISCU2}$, and a water and after Cys addition and Cys381$^{NFS1}$ persulfidation is ready for sulfur transfer. Finally, FDX2-dependent reduction of the persulfide to inorganic sulfide and transient dimerization of two [1Fe-1S]-bound ISCU2 molecules enables [2Fe-2S] cluster formation[18].

The model depicted in Fig. 5 presents the various conformational states of the core ISC complex defined in this work. The entire process starts with the assembly of Fe-bound ISCU2 and (NIA)$_2$ to form the (Fe-NIAU)$_2$ complex (Fig. 5, stages 1 and 2). According to Mössbauer- and CD-spectroscopic analyses, free ISCU2 binds ferrous Fe in a tetrahedral coordination via Cys69, Asp71, Cys95 and His137 similar to mouse Fe-U[30] (PDB 1WFZ). Mössbauer data suggest that a similar Fe species is present in a major fraction of the (Fe-NIAU)$_2$ sample, yet structural data for both (Fe-NIAU)$_2$ (this work) and (Zn-NIAU)$_2$[15] indicate that in these complexes Cys69$^{ISCU2}$ is replaced by Cys381$^{NFS1}$ and His137$^{ISCU2}$ by a water (Fig. 5, stage 2). Such a conformational change is, however, Mössbauer-silent (component 1 in Table 3). The Mössbauer technique additionally revealed the presence of another (minor) species with distinct parameters (component 2). Addition of both Cys and FXN substantially increased the amount of this component 2 indicating an equilibrium between (at least) two states (Fig. 5, stages 3 and 4, 5). As FXN does not stably and quantitatively bind to (NIAU)$_2$ due to low affinity, the residual amount of component 1 likely represents (Fe-NIAU)$_2$ lacking FXN, while the FXN-containing complexes (i.e., (Fe-N$^S$IAUX)$_2$ sulfur transfer intermediate of stage 4 and (Fe-NIAU$^S$X)$_2$ of stage 5) may feature component 2. In vivo, it is well possible that the (NIA)$_2$ complex already contains a persulfidated NFS1 when binding to Fe-ISCU2, thus creating stage 3 without a transient stage 2. Nevertheless, our experiments suggest that also stage 2, upon addition of Cys, can be readily converted to stage 3, possibly due to the high dynamics of the Cys-loop of NFS1.

The cryo-EM structures suggest a highly conformationally dynamic ISCU2 assembly site upon transition from (Fe-NIAU)$_2$ to persulfidated and FXN-bound states involving dissociation of Cys381$^{NFS1}$ from the Fe and binding of Cys138$^{ISCU2}$ (Fig. 5, stages 3–5). Since our persulfide-containing cryo-EM structures (Fe-N$^S$IAU$^S$X)$_2$ and (Fe-

NIAU$^S$X)$_2$ likely represent pre- and post-sulfur transfer states, the conformational flexibility at the ISCU2 assembly site may be central for bringing the sulfur-donating Cys381$^{NFS1}$ and sulfur-accepting Cys138$^{ISCU2}$ in close vicinity (from 6.5 Å in (NIAU)$_2$ to 3.6 Å in (Fe-N$^S$IAU$^S$X)$_2$; Table 1) to facilitate sulfur transfer. This distance may be sufficient for direct transfer, considering the S–S bond length of a persulfide of 2.0 Å[38]. FXN, even though not essential, apparently shifts the equilibrium of these states and thereby accelerates the structural transitions toward the configuration that is optimal for sulfur transfer. This action may be represented by the (Fe-N$^S$IAU$^S$X)$_2$ structure. It is not clear whether the unexpected detection of two persulfide sulfurs in this structure represents a true reaction intermediate or is explained by the presence of particles in both pre- and post-sulfur transfer states that were averaged during cryo-EM analysis. However, the vicinity of Cys381$^{NFS1}$ and Cys138$^{ISCU2}$ at the coordinating Fe provides a chemically favorable orientation for facile sulfur exchange by nucleophilic attack.

Since only the conformation stabilized by FXN but not that of (Fe-NIAU)$_2$ (or component 1) sterically allows the persulfide transfer, this structural transition likely defines the molecular mechanistic role of FXN in the process. Precisely, FXN interaction with persulfidated (Fe-N$^S$IAU)$_2$ leads to movements of (1) His137$^{ISCU2}$ into a (hydrophobic) pocket formed by Pro163$^{FXN}$ and Trp155$^{FXN}$ and (2) Cys69$^{ISCU2}$ further away from the assembly site, thereby (3) closing ranks between Fe and Cys138$^{ISCU2}$ (Fig. 2a and Supplementary Movie 1). The resulting closer vicinity of sulfur-donating Cys381$^{NFS1}$ and sulfur-accepting Cys138$^{ISCU2}$ and the exit of sterically hindering His137$^{ISCU2}$ from the assembly site facilitates the sulfur transfer reaction. Interestingly, this conformational transition can also occur, yet at much slower rate, without FXN, as seen in our persulfidation experiments (Fig. 3d), suggesting that FXN simply shifts the equilibrium between these states. Structurally, a similar transition is observed in the persulfide-free and Zn-containing

$(NIAUX)_2$ complex, despite the presence of non-physiological Zn which supports sulfur transfer but not [2Fe-2S] cluster formation[15,17,30]. To finish the biosynthetic process, persulfide reduction by FDX2 enables the formation of mature holo-ISCU2 (Fig. 5; [2Fe-2S]-U)[18, 20].

For detailed analysis of the persulfidation reaction, we have developed an assay employing iodoTMT labels with different masses to quantitatively determine persulfidation of all involved Cys residues in parallel under anaerobic conditions. Using this assay, we show a direct, rapid, and specific persulfide transfer from $Cys381^{NFS1}$ to $Cys138^{ISCU2}$. By slowing down the reaction at 0 °C and by using short reaction times (1 s), we could make other Cys persulfides as intermediates unlikely, even though sterically $Cys95^{ISCU2}$ could serve as a sulfur acceptor (see below). Our persulfidation experiments further resolved the potential roles of the other active-site residues in sulfur transfer. While $Asp71^{ISCU2}$ and $Cys95^{ISCU2}$ are essential for persulfidation because of their continuous Fe coordination, surprisingly both $Cys69^{ISCU2}$ and $His137^{ISCU2}$ were not. Since the latter two residues coordinate the Fe in free ISCU2, Fe must be able to enter the process also at a later stage, e.g., at the $(NIAU)_2$ complex, thus making a free Fe-U complex dispensable. The non-essential role of $Cys69^{ISCU2}$ for persulfidation is fully consistent with its structural distance to the $Cys381^{NFS1}$ persulfide after $(Fe-N^SIAU)_2$ formation and later steps. Together, these results suggest a non-essential role of $Cys69^{ISCU2}$ and $His137^{ISCU2}$ during persulfide transfer.

Concerning the mechanism of persulfidation, our structural data showing similar distances of $Cys381^{NFS1}$ persulfide to both $Cys95^{ISCU2}$ and $Cys138^{ISCU2}$ (Table 1) did not readily exclude a sulfur relay to $Cys95^{ISCU2}$. With a CSS bond angle of around 103°[39], both $Cys95^{ISCU2}$ and $Cys138^{ISCU2}$ would be similarly positioned to accept the sulfur from persulfidated $Cys381^{NFS1}$. To detect such a possible (intermediate) species, we adapted our assay conditions so that the fast time course of $Cys138^{ISCU2}$ persulfidation could be temporally resolved. In these experiments at 0 °C and short reaction times, exclusive persulfidation of $Cys138^{ISCU2}$ was observed, making a $Cys95^{ISCU2}$ intermediate unlikely (Fig. 3e). Consistently, no persulfidation of other Cys residues was detectable for the ISCU2-C138S variant. These results raise the interesting question of why $Cys138^{ISCU2}$ is preferred over $Cys95^{ISCU2}$. One possible explanation could be that $Cys95^{ISCU2}$, together with $Asp71^{ISCU2}$, continuously coordinates the Fe throughout the pathway, and thus may not be flexible enough to accept an extra atom. In contrast, $Cys138^{ISCU2}$ shows comparatively high conformational flexibility, e.g., during the movement toward the Fe upon the FXN-accelerated conformational change of ISCU2 (Supplementary Movie 1). Interestingly, persulfide transfer from NFS1 to ISCU2 observed here (Fig. 3) appears to be much faster than previously assumed based on measurements from 10 s to several hours[25,26,30]. This high rate was even exceeded by NFS1 persulfidation which at 0 °C could not kinetically be resolved (Supplementary Fig. 11d) and may occur at similar fast rates as the formation of the PLP-intermediates[26]. Hence, in vivo, $(NIA)_2$ may be persulfidated before binding its partner proteins, and persulfidation of $Cys381^{NFS1}$ only after $(Fe-NIAUX)_2$ complex assembly seems kinetically unlikely (Fig. 5, stage A). Overall, our study suggests that the persulfidation steps are not rate-limiting for the overall process of [2Fe-2S] cluster synthesis.

Our work delineates a potential mechanism of persulfide transfer within the core ISC complex and defines the molecular role of FXN in this process. As indicated by our study, the molecular transitions facilitated by FXN occur at low efficiency in the absence of this protein, and may serve as the basis for the development of small molecules mimicking FXN function for therapeutic regimes of Friedreich's ataxia. A further important objective for future investigations is the structural characterization of intermediates upon and after FDX2 reduction of $(Fe-NIAU^SX)_2$ to yield a putatively [1Fe-1S]-containing complex. The presumably short-lived nature of such intermediates may render their characterization rather challenging.

## Methods

### Expression and purification of recombinant proteins

An overview of the plasmid constructs used for recombinant protein expression is given in Supplementary Table 3. FDXR[22] and all other proteins[20] were overproduced and purified at 4 °C or on ice as described hereinafter. His-tagged constructs were purified by His-affinity chromatography and nontagged constructs by anion exchange chromatography (AEC). Pellets were resuspended in His or AEC buffers (respective protein-specific buffer composition given in Supplementary Table 4). Protease inhibitor (cOmplete Protease Inhibitor Cocktail), lysozyme and DNase I (and PLP for NIA purification) were added to the suspension. Cells were lysed by sonication and the lysate cleared by centrifugation. The cell extract was loaded onto either a His-binding (Ni-NTA-Agarose) or an anion exchange column (Source 30Q) pre-equilibrated with His or AEC buffers. His-binding columns were washed using His buffer plus 10 mM $((NIA)_2)$ or 70 mM (ISCU2, FXN) imidazole and proteins eluted in His buffer plus 250 mM imidazole. Anion exchange columns were washed with AEC buffer, and proteins were eluted using a gradient of 0–100% AEC elution buffer (Supplementary Table 4). Eluates were concentrated to a volume of ca. 1.5 ml. FXN was treated with recombinant TEV protease to remove the N-terminal His-tag. Proteins were transferred to an anaerobic chamber and $(NIA)_2$, ISCU2 and FXN incubated for 60 min with additives (Supplementary Table 5) to remove metals and/or polysulfanes/sulfides. Protein samples were subjected to anaerobic size exclusion chromatography using HiLoad 16/600 Superdex columns pre-equilibrated with degassed SEC buffer (Supplementary Table 4). Desired fractions were combined, concentrated to roughly 1 ml, aliquoted in air-tight glass vials and checked for purity by SDS–PAGE. Protein concentration was determined via the Bradford assay. A correction factor for the resulting concentrations was determined for $(NIA)_2$, ISCU2, FXN, and FDX2 by quantitative amino acid analysis (Leibniz-Institut für Analytische Wissenschaften).

Purification of $(NIA)_2$ typically consisted of His affinity chromatography followed by SEC. For cryo-EM sample preparation, an AEC step was added after His purification to achieve the highest possible purity. ISCU2-D71A mutant protein showed aggregate formation and Fe/S cluster binding, when purified under the same conditions employed for wild-type ISCU2 and other ISCU2 variants. To maintain ISCU2-D71A in a monomeric apoform, buffer and additive compositions were optimized as detailed in Supplementary Tables 4 and 5.

### Cryo-EM sample preparation

Protein preparations of high purity were used for cryo-EM experiments (85% purity of $(NIA)_2$ and 92–99% purity of all other proteins, based on densitometry of SDS-PAGE bands). Cryo-EM sample preparation was performed at room temperature under oxygen-free conditions in an anaerobic chamber (Coy Laboratories) using anaerobic buffers. $(NIA)_2$ and ISCU2 were mixed in a 1:2 molar ratio and incubated for 5 min. The formed $(NIAU)_2$ complex was purified by SEC using a Superdex 200 increase 10/300 GL column (GE Healthcare) equilibrated with cryo-EM buffer (35 mM Tris/HCl pH 7.4, 150 mM NaCl) and concentrated using a 100 kDa centrifugal filter (10 min, $4000 \times g$).

UltrAuFoil 0.6/1 300 mesh gold grids (Quantifoil) were made hydrophilic by plasma cleaning using a NanoClean Model 1070 (Fischione Instruments) running for 2 min at 40% power with an $Ar/O_2$ gas mixture (90:10 ratio) prior to sample vitrification. 27.5 μM $(NIA)_2$, 275 μM Na-ascorbate, 275 μM $FeCl_2$ and 55 μM FXN were mixed in cryo-EM buffer and the persulfidation reaction was initiated by addition of 300 μM cysteine. The sample (2.7 μl) was mixed with 0.3 μl fluorinated fos-choline-8 (Anatrace; final concentration 1.5 mM), and applied to the freshly plasma-cleaned grid. The sample was vitrified in liquid ethane using a Vitrobot Mark IV (Thermo Scientific) inside the anaerobic chamber. The grid was blotted for 4 s with a nominal blot force of 4 at 4 °C and 100% humidity using grade 595 filter paper (Whatman

Products). The time between adding cysteine and sample vitrification was less than 1 min.

## Data acquisition and image processing

Cryo-EM data were acquired in Energy-Filtered Transmission Electron Microscopy (EFTEM) mode using a Titan Krios G3i electron microscope (Thermo Scientific), equipped with a BioQuantum energy filter (Gatan) and operated at 300 kV. Electron-optical alignments were adjusted with EPU (Thermo Scientific). In total, 5410 dose-fractionated movies were recorded with a K3 direct electron detector (Gatan) in electron counting mode, using aberration-free image shift (AFIS) automation strategies of EPU, with a total dose of ~62 e⁻/Å² spread equally over 60 frames at a nominal magnification of ×105,000, corresponding to a calibrated pixel size of 0.837 Å.

Data quality was monitored during data acquisition using cryoS-PARC live[40]. In total, 4298 micrographs with a CTF estimation better than 5 Å and astigmatism less than 500 Å were used for initial data processing in cryoSPARC as described hereafter (Supplementary Fig. 1a). Particles were picked with the blob-picker implementation and subjected to 2D classification. 2D classes showing the $(NIAUX)_2$ complex were selected and used for template picking. Initially, 2,189,680 particles were picked, extracted with a box size of 256 × 256 pixels and classified in 2D. Based on the appearance of 2D class averages, a subset of 1,075,763 particles was subjected to a homogeneous refinement in cryoSPARC resulting in a global resolution of 2.9 Å. Particle coordinates from this refinement were then converted into a star file using Pyem[41] for further data processing in RELION-3.1[42].

Movies were sorted into optics groups according to their EPU AFIS metadata (https://github.com/DustinMorado/EPU_group_AFIS). Full-frame motion correction was performed with MotionCor2[43] using 3 × 3 patches and the contrast transfer function (CTF) was calculated using CTFFind4.1[44]. Only micrographs with a CTF estimated better than 5 Å were used (4508 micrographs). Particle coordinates initially obtained in cryoSPARC live were extracted in RELION-3.1 with a box size of 256 × 256 pixels and 4x down sampling (1,003,002 particles) and subjected to 2D classification. A subset of 483,881 particles was selected on the basis of high-quality 2D classes and used to generate an initial model. A 3D classification (25 iterations, T = 4) of all 1,003,002 particles with this initial model as 3D reference (60 Å low pass filtered) resulted in a major class containing 696,662 particles, which were used for further 3D refinements. After re-extraction of the refined particles using 1.2x down sampling, a 3D refinement applying C2 symmetry gave a global resolution of 2.86 Å (0.143 FSC criterion). Two rounds of CTF refinement, correcting for higher order aberrations, anisotropic magnification and per-particle defocus, with every optics group being refined independently[45], followed by Bayesian particle polishing, further improved the overall resolution to 2.41 Å. Particles were C2 symmetry expanded and local 3D classification without alignment (T = 4, 50 iterations) was performed with a mask focusing on the FXN-binding region (Supplementary Fig. 1a). This separated the dataset into 2 classes−FXN-bound (77.1%) and non-FXN-bound (22.9%), which were refined by local 3D refinements using a solvent mask yielding resolutions of 2.41 Å and 2.71 Å, respectively.

The non-FXN-bound subset was further separated by a focused 3D classification (40 iterations, T = 16, without alignment) applying a mask on ISCU2. The major class contained 269,101 particles, which were subjected to a local 3D refinement (2.75 Å final resolution). Map interpretability could be further improved by density modification[46] and local anisotropic sharpening[47] using the Phenix software package[48].

The FXN-bound map showed density consistent with a persulfide on Cys138ISCU2 and revealed two alternate conformations of the flexible Cys loop of NFS1. To separate these loop conformations a multi-reference 3D classification approach was used. References were generated from a model based on PDB 6NZU, which was real-space refined against the FXN-bound cryo-EM map followed by manual fitting of two versions of the Cys loop of NFS1 (residues 377-389) into density corresponding to the two alternate conformations using Coot-0.9[49] (Supplementary Fig. 1a). 3D volumes, low-pass filtered to 8 Å, were generated from these models using the *molmap* command in ChimeraX[50]. Particle subtraction with a mask focusing on the Cys loop region was performed and multi-reference 3D classification (T = 64, 25 iterations, without alignment) was carried out using 10 Å low-pass filtered 3D references to separate the alternate loop conformations. Particles from both classes were subjected to another round of multi-reference classification using the same parameters and references. Particles assigned to the major classes were selected, reverted to the original (non-signal subtracted) particles and subjected to a local 3D refinement using the initial model as a 3D reference. Conformation 1 (inward), resolved to a global resolution of 2.49 Å, shows Cys381NFS1 near the PLP cofactor, while in conformation 2 (outward), resolved to 2.58 Å, Cys381NFS1 is close to the ISCU2-bound Fe. Density modification[46] and local anisotropic sharpening[47] in Phenix[48] further improved map clarity.

For the electron dose analysis (Supplementary Fig. 4), movie frames 1–2 (dose 2.07 e⁻ Å⁻²), 1–3 (dose 3.1 e⁻ Å⁻²), 1–5 (dose 5.17 e⁻ Å⁻²), 1–10 (dose 10.33 e⁻ Å⁻²), 1–40 (dose 41.33 e⁻ Å⁻²) and 1–60 (dose 62 e⁻ Å⁻²) were combined during Bayesian particle polishing in RELION-3.1. Half-maps were reconstructed with *relion_reconstruct* and post-processed using the *relion_postprocess* command with automatically estimated B-factor.

## Model building and refinement

The cryo-EM structure of human $(Zn-NIAUX)_2$ (PDB 6NZU)[17] was used as a template for model building and refinement. The model was rigid-body fitted into the respective cryo-EM map in ChimeraX followed by manual refinement in Coot-0.9[49]. Persulfide sulfur (S-mercaptocysteine; CSS) and Fe²⁺ (FE2) were added to the model in Coot-0.9. Metal restraints were generated from the manually refined model using ReadySet! in Phenix, and adjusted according to published high-resolution structures. Water molecules were built manually in agreement with the density and the hydrogen-bonding distances. The final models were real-space refined against the respective map using *phenix.real_space_refine* and validated by *phenix.validation_cryoem* within Phenix[48] (Table 2). Figures and the movie of models and maps were prepared with ChimeraX[50].

## Enzyme assays, CD spectroscopy and analytical SEC

All persulfide transfer and reconstitution assays, analytical SEC (aSEC) as well as sample preparation for circular-dichroism (CD) spectroscopy were performed under oxygen-free conditions in an anaerobic chamber (Coy Laboratory products) using anaerobic buffers. aSEC was performed on an UltiMate 3000 UHPLC system (Thermo Fisher Diagnostics) using a Superdex 200 increase 10/300 GL column (GE Healthcare) equilibrated with anaerobic SEC buffer (Supplementary Table 4). CD spectroscopy was performed anaerobically at room temperature in a quartz cuvette with 4 mm path length sealed with an air-tight rubber capping on a Jasco J-815 (JASCO Inc.).

## Reconstitution of Fe-ISCU2

Fifty μM ISCU2 was incubated in reconstitution buffer (35 mM Tris/HCl pH 8.0, 150 mM NaCl, 5% w/v glycerol, 2 mM TCEP) for 5 min at room temperature. CD spectra (250−350 nm) were recorded before and after addition of 2 eq. $FeCl_2$ per ISCU2. The apo-ISCU2 spectrum was subtracted from the Fe-ISCU2 spectrum to remove signal contributions from aromatic amino acids[51].

## Persulfidation assay with MPB-based persulfide quantification

NFS1-dependent persulfidation of ISCU2 was analyzed based on Cys alkylation[25]. Standard reactions contained 100 μM $FeCl_2$, 100 μM Na-

ascorbate, 20 μM ISCU2, 40 μM FXN and 20 μM $(NIA)_2$ in MPB buffer (35 mM Tris/HCl pH 7.4, 150 mM NaCl, 5% w/v glycerol). After incubation for 3 min at room temperature, 200 μM cysteine was added to initiate persulfide formation, and the reaction was quenched after 10 s by mixing with 2–8 eq. (per total thiol concentration) of maleimidepolyethylenglycol₁₁-biotin (MPB) dissolved in DMSO. Directly afterwards, 1% w/v SDS was added, and samples were mixed vigorously and incubated for 10 min. Aliquots containing 1 μg ISCU2 were mixed with SDS-PAGE sample buffer containing 50 mM TCEP and incubated at room temperature for 15 min. The reduced samples were analyzed either by standard SDS-PAGE using 15% w/v acrylamide gels (for ISCU2 WT and variants with at least three Cys residues remaining) or by tricine SDS-PAGE using 16% w/v acrylamide gels (ISCU2 variants with fewer than three Cys residues). Gels were stained with InstantBlue (Expedeon), and ISCU2 bands were quantified by densitometry using Image Studio Lite version 5.2 to calculate the percentage of persulfidated ISCU2-species.

### Persulfidation assay with iodoTMT-based persulfide quantification

Persulfidation of individual Cys residues of NFS1 and ISCU2 was quantified via mass spectrometric analysis of persulfide transfer reactions labeled with thiol-specific iodoacetyl Tandem Mass Tag™ (iodoTMT) reagents. Stocks of 22.1–30.0 mM iodoTMT dissolved in 50% v/v methanol, 5–8% w/v SDS, 3–5 mM Na-EDTA and 100 mM Tris pH 8.5 were prepared directly before the experiment. Reactions contained 100–150 μM $FeCl_2$ (5 eq. over ISCU2), 20–30 μM ISCU2, 20–40 μM FXN and 15–40 μM NIA in iodoTMT buffer (35 mM Tris/HCl pH 8.0, 150 mM NaCl, 5% w/v glycerol). Following incubation for 3 min at room temperature, 75–200 μM cysteine (2.5 eq. over $(NIA)_2$) was added to initiate persulfide formation. Reactions were quenched at time points up to 320 s with 4.4–15.0 mM iodoTMT (10.3–39.5 eq. over total thiols) and incubated for 10 min. Samples were incubated with 6.2–22.0 mM cysteine (1.5 eq. over remaining iodoTMT) for 10 min to quench the labeling reagent, and persulfides were cleaved by incubation with 5 mM TCEP for 10 min. Cys residues were subsequently labeled for 10 min with 4.4–15 mM (2.0 eq. over remaining cysteine) of a second iodoTMT reagent of different reporter fragment mass. Remaining labeling reagent was quenched with 6.2–15 mM cysteine for 10 min. Samples were digested by the addition of Sequencing Grade Modified Trypsin (Serva) and incubated at 37 °C overnight. Peptides were not reduced or carbamidomethylated because of iodoTMT-labels. After tryptic digestion, resulting peptides were desalted and concentrated using Chromabond C18WP spin columns (Macherey-Nagel, Part No. 730522). Finally, peptides were dissolved in 25 μl of water with 5% acetonitrile and 0.1% formic acid.

The mass spectrometric analysis of the samples was performed using either an Orbitrap Velos Pro mass spectrometer or an Orbitrap Exploris 480 (both Thermo Scientific). Both systems were connected to Ultimate nanoRSLC-HPLC systems (Thermo Scientific), equipped with custom end-fritted 50 cm × 75 μm C18 RP columns, filled with 2.4 μm beads (Dr. Maisch) and connected online to the mass spectrometers through nanospray sources. Two μl, corresponding to 2 μg, of the tryptic digests were injected onto a 300 μm ID × 1 cm C18 PepMap pre-concentration column (Thermo Scientific). Automated trapping and desalting of the sample were performed at a flowrate of 6 μl/min using water/0.05% formic acid as solvent. Separation of the tryptic peptides was achieved using a gradient of water/0.05% formic acid (solvent A) and 80% acetonitrile/0.045% formic acid (solvent B) at a flow rate of 300 nl/min, holding 4% B for 5 min, followed by a linear gradient to 45% B within 30 min and linear increase to 95% solvent B for further 5 min.

On the Orbitrap Velos Pro system, a survey scan with a resolution of 60,000 within the Orbitrap mass analyzer was combined with targeted FT-$MS^2$ scans (resolution of 15,000) using HCD fragmentation (charge state adapted, normalized CE 35%). Ionization parameters were as follows: Capillary temperature 270 °C, source voltage 2.7 kV, S-lens RF-level 64.6. On the Orbitrap Exploris 480, a resolution of 60,000 was used for the survey scans and of 45,000 for targeted $MS^2$ scans. A fixed scan range of 120–1200 $m/z$ was used for the targeted $MS^2$ scans to ensure that the reporter ions were properly measured. Ionization parameters were as follows: Temperature of ion transfer tube 275 °C, source voltage 2.3 kV. On both mass spectrometers, targeted $MS^2$-spectra were recorded for the following $m/z$ of the targets: ISCU2: 959.5085, 748.7231, 614.3708, 830.4421; NFS1: 485.2727, 447.7556, 815.4144, 848.1060. Extracted ion chromatograms corresponding to the reporter fragment ion masses were generated from $MS^2$ data scans with 5 ppm (Orbitrap Velos Pro) or 10 ppm (Orbitrap Exploris 480) mass accuracy using the QualBrowser of the Xcalibur software package (Thermo Scientific). The resulting signals were manually integrated for quantification purposes (Supplementary Fig. 13). For each protein Cys residue, the percentage of persulfidation was calculated from the ratio of detected amounts of both iodoTMT labels.

### Mössbauer spectroscopy

Samples were prepared under strictly anaerobic conditions in an anaerobic chamber (Coy). Two sets of Mössbauer samples were prepared. The first set (Supplementary Fig. 12 and Supplementary Table 1) was prepared employing ⁵⁷Fe-enriched ferric ammonium citrate (FAC). Chemicals listed in Supplementary Table 2b were mixed in buffer 1 (50 mM Tris, 150 mM NaCl, 5% w/v glycerol, pH 8.0) in the given order. Subsequently, a buffer exchange into buffer E (buffer 1 plus 2 mM TCEP) was performed using ZebaSpin 7 K columns immediately before flash freezing. Resulting samples exhibited significant amounts of non-ISCU2-specific Fe species (components A and B) as impurities (Supplementary Table 1). The second set was prepared under optimized conditions (Supplementary Table 2a) that drastically decreased these Fe impurities (Fig. 4 and Table 3). In this case, $(NH_4)_2$⁵⁷$Fe(SO_4)_2$ was synthesized from elemental ⁵⁷Fe and a mixture of $(NH_4)_2SO_4$ and $H_2SO_4$[52]. Chemicals listed in Supplementary Table 2a were mixed with buffer 1 (50 mM Tris, 150 mM NaCl, 70% w/v glycerol, pH 8.0) to achieve a final glycerol content of 30% w/v at a final volume of 360 μl. Buffers, ISCU2, $(NH_4)_2$⁵⁷$FeSO_4$ and Na-ascorbate were mixed first, and samples were incubated for 5–10 min. Cysteine was added last to samples to initiate persulfide formation. Directly afterwards, 240 μl were pipetted into Mössbauer cups. Samples were flash-frozen in liquid nitrogen inside the anaerobic chamber. This resulted in an incubation time with Cys of ~2 min. The [2Fe-2S]-loaded ISCU2 sample was incubated for 40 min before flash freezing, and the formation of the [2Fe-2S] cluster was confirmed by analyzing an aliquot (50 μl) by CD spectroscopy.

Mössbauer transmission data of the prepared samples were recorded and processed using a conventional Mössbauer spectrometer in constant acceleration mode. The setup included a 512-channel analyzer (WissEl GmbH) in the time-scale mode. Calibration of spectra was performed at room temperature with α-Fe foil. Zero-field spectra were recorded upon sample cooling to 77 K via a continuous flow cryostat (Oxford Instruments) filled with liquid nitrogen. Data analysis was performed using the Microsoft Excel add-on VindaB[53], employing least-squares fits with Lorentzian line shapes. The relative transmissions given as data points in Fig. 4 and Supplementary Fig. 12 were calculated by dividing the registered total counts of each channel by the spectral background.

### Reporting summary

Further information on research design is available in the Nature Portfolio Reporting Summary linked to this article.

## Data availability

Cryo-EM maps have been deposited in the Electron Microscopy Data Bank (EMDB) under the following accession codes: EMD-17731

((Fe-NIAUX)$_2$ consensus map), EMD-17732 ((Fe-NIAU$^S$X)$_2$ structure), EMD-17733 ((Fe-N$^S$IAU$^S$X)$_2$ structure), and EMD-17734 ((Fe-NIAU)$_2$ structure). Atomic models have been deposited in the Protein Data Bank (PDB) under the following accession codes: 8PK8 (Fe-NIAU$^S$X)$_2$ structure), 8PK9 ((Fe-N$^S$IAU$^S$X)$_2$ structure), and 8PKA, ((Fe-NIAU)$_2$ structure). Previously published structures used in this study are available in the PDB with the accession codes 1WFZ (*M. musculus* IscU solution NMR structure), 5WLW (*H. sapiens* (Zn-NIAU)$_2$ X-ray structure) and 6NZU (*H. sapiens* (Zn-NIAUX)$_2$ cryo-EM structure). Any additional information required to reanalyze the data reported in this study is available from the corresponding authors. Source data are provided with this paper.

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

## Acknowledgements

We acknowledge the contribution of the Core Facilities "Mass spectrometry and Elemental analysis", "Protein Biochemistry and Spectroscopy" of the Philipps-Universität Marburg, and the "Central Electron Microscopy Facility" of the Max Planck Institute of Biophysics in Frankfurt. We thank Dr. Timo Glatter (MPI Marburg) for letting us use the Exploris 480 mass spectrometer. R.L. acknowledges generous financial support from Deutsche Forschungsgemeinschaft (Koselleck grant LI 415/6, and SPP 1927, LI 415/7), B.J.M. from the Max Planck Society, and Vo.S. from Deutsche Forschungsgemeinschaft (SPP 1927, SCHU 1251/17-2).

## Author contributions

Conceptualization was by Vo.S., B.J.M., and R.L. V.S. and S.A.F. prepared the protein samples for cryo-EM, biochemistry, and Mössbauer spectroscopy. R.S. prepared the cryo-EM grids, acquired and processed cryo-EM data, and built the atomic models. S.W. aligned the electron microscope and supported data acquisition. R.S. and B.J.M. analyzed and interpreted the cryo-EM data. V.S., S.A.F., and N.K. performed the biochemical experiments and analyzed them with R.L. U.L. performed the mass spectrometric analyses including the iodoTMT experiments which were evaluated with V.S. J.O. and M.H.H. acquired the Mössbauer data which were evaluated together with Vo.S. V.S., R.S., and J.O. prepared the figures and tables. Writing of the original draft was by V.S., R.S., Vo.S., B.J.M., and R.L. Review and editing of the manuscript was by all authors. Visualization was by V.S., R.S., J.O., S.A.F., and R.L. Supervision and funding acquisition was by Vo.S., B.J.M., and R.L.

## Funding

## Competing interests

The authors declare no competing interests.
