## [Peer Review File · Nature Communications]

Mechanism and structural dynamics of sulfur transfer during de novo [2Fe-2S] cluster assembly on ISCU2REVIEWERS' COMMENTS

Reviewer #1 (Remarks to the Author):

The current manuscript is a continuation of a previous work that propose a molecular mechanism for the mitochondrial [2Fe-2S] cluster biogenesis, focused on studies defined the role of Tyr35 in the de novo synthesis.

Synthesis of iron-sulfur (Fe/S) clusters (ICS) in living cells requires scaffold proteins for both facile synthesis and subsequent transfer of clusters to target apoproteins. The (ISC) complex is formed by the dimeric pyridoxal phosphate (PLP)-dependent cysteine desulfurase NFS1 and two bound regulatory proteins ISD11 and ACP, and firstly generates an enzyme-bound persulfide (-SSH) from free cysteine. The ISC protein frataxin (FXN) allosterically activates persulfide transfer from NFS1 to one of the three conserved Cys residues of iron-containing ISCU2, binding to opposite tips of the NFS1 dimer. Persulfide reduction by the ferredoxin FDX2 (and its reductase FDXR) generates the sulfide required for [2Fe-2S] cluster formation. These initial steps have been investigated before but the molecular mechanisms underlying the actual [2Fe-2S] cluster synthesis was still unknown.

In this new manuscript, the authors investigated the Fe-dependent mechanism of per sulfide transfer from cysteine desulfurase NFS1 to ISCU2. They used high-resolution cryo-EM structures to prove the persulfide transfer from Cys381NFS1 to Cys138ISCU2 and explained the molecular role of frataxin in the sulfur transfer.

The manuscript is well written and the experiments are presented in a logic and clear sequence. The work defines for the first time the molecular role of FXN in this process. Interestingly, the molecular transitions facilitated by FXN occur with low efficiency in the absence of frataxin. It may serve as the basis for the development of small molecules mimicking FXN function for therapeutic treatments of Friedreich's ataxia

The work presented here is of high significance in the field, considering that is an extra step in the direction of understanding more clearly, what the frataxin role is.

The methodology used is sound and the conclusions made are justified by the data. I admire the clever use of persulfide transfer reactions labelled with thiol-specific iodoacetyl Tandem

746 Mass Tag™ (iodoTMT) reagents for the HRMS experiments.

Several minor points:

1. In the Experiment “Persulfidation assay with iodoTMT-based persulfide quantification”, what will happen if instead of “100 – 150 μM FeCl_2 ”, the authors would add ZnCl_2 ?
2. More details would be helpful to replicate the LC-MS methodology for the source parameters for both Orbitraps.
3. What was the amino acids sequence of the frataxin used in the experiments?
4. Can the MS2 spectra be added in the extended figures?
5. I recommend one more reading for some words that used a synonym instead of the regular English term

Reviewer #2 (Remarks to the Author):

This manuscript by Schulz et al. sheds new mechanistic light on the ISC complex, which catalyzes [2Fe-2S] cluster assembly as a critical step in the biogenesis of mitochondrial Fe/S proteins. In the early steps of the reaction, Cys381 of NFS1 receives persulfide from the desulfuration of cysteine by (NIA)2. The final acceptor of this persulfide is Cys138 of ISCU2, but the mechanism of persulfide transfer between these subunits, as well as the interplay of interactions involving Fe and persulfide during the reaction, was largely unknown. Prior work performed with Zn- rather than Fe-containing complexes left significant unanswered questions, which the authors have attempted to address here through careful structural characterization of iron-containing ISCU2/NFS1/ISD11/ACP1/FXN complexes prepared under anaerobic conditions. Creative and rigorous cryo-EM data processing allowed the authors to identify three distinct particle sub-populations that were each reconstructed at high resolutions, ranging from 2.4 to 2.7 Å. These sub-populations comprised both frataxin-bound and frataxin-free states, and within the frataxin-bound state, two alternate conformations of the NFS1 Cys-loop were distinguished, which helped to elucidate the mechanism of sulfur transfer and the role of frataxin in promoting the reaction, as well as to clarify the specific cysteine residues that participate in the reaction. These mechanistic hypotheses were further probed through careful biochemical and spectroscopic experiments. The structural work reported here is of high technical quality, particularly considering the requirement for work under anaerobic conditions, and the conclusions represent a significant conceptual advance in the field that justifies its publication in Nature Communications.

That said, I have a few relatively minor comments and questions.

1. Given the delicate nature of the sulfur-based structures to be resolved in this study, is there a reason the authors elected to use a fairly high total electron dose ($60 \text{ e}^-/\text{\AA}^2$) during data collection? This might be expected to run the risk of breaking relatively labile S-S bonds.
2. How do the authors account for the observed structural differences between their Fe-bound structures and previously published Zn-bound structures? Do the varying atomic distances reflect

different transient intermediates of the complex during the reaction or do the authors think the presence of Zn²⁺ may introduce structural artifacts?

3. Sample and reaction preparation were performed under anaerobic conditions. Was sample vitrification similarly maintained in anaerobic conditions?

4. What was the reason for the 1.2x downsampling of the particles in the final reconstructions?

5. In figure 1, It would be nice to see a more convincing representation of the map density that provided evidence for persulfidated cysteines.

Reviewer #3 (Remarks to the Author):

Review for manuscript NCOMMS-23-53565-T

The manuscript presented by V. Schulz and colleagues describes a study on the role of frataxin in facilitating persulfide transfer between the cysteine desulfurase NFS1 and the scaffold protein ISCU. To achieve this, the authors combined structural analyses using cryo-electron microscopy with solution-based approaches, including mass spectrometry, titrations and labeling analyses, as well as Mossbauer spectroscopy analysis.

In a very elegant manner, the authors successfully isolated particles of the (Fe-NIAUX)₂ complex at various stages during the persulfide transfer between the active site of NFS1 and the iron-sulfur cluster assembly site. Using cutting-edge cryo-electron microscopy approaches, they were able to discriminate between different types of particles, obtaining structures with a resolution of around 2.5 Å with and without frataxin. Importantly, these structures revealed a ferrous ion bound in the assembly site and a persulfide located either on the NFS1 loop (pre-transfer) or on ISCU (post-transfer). The combination of these different structures allows the authors to propose a scenario of events leading to this persulfide transfer. Remarkably, the authors demonstrate, for the first time, that the ferrous ion is bound to the protein in a manner different from what had been proposed based on zinc-binding structures. Moreover, they illustrate how frataxin enables a local structural rearrangement to facilitate a direct persulfide transfer, thereby shedding light on the initial steps of molecular-level iron-sulfur cluster formation. This structural work is of outstanding quality.

The second part of the manuscript corresponds to in vitro analyses necessary to support the structural work. Collectively, these data match the structural observation and clearly support the proposed mechanism. The reviewer is not an expert in Mossbauer spectroscopy and cannot truly assess the quality of the interpretation. However, the authors seem to assume that the signals observed by this method necessarily correspond to the species observed by cryo-electron microscopy. For instance, the authors consider that the signal they term 'component 2' is necessarily associated with the structure observed by cryo-EM. However, the sample concentrations between the two types of experiments are quite different, and one cannot exclude that at such high concentrations used for the Mossbauer

spectroscopy study, a significant amount of iron could be released, and the signal corresponding to an isomer shift of 1.1 – 1.15 mm/s could be that of this free iron.

Minor points:

Regarding the observation (or at least the presentation) of a structure of the NIAUX complex with two persulfides simultaneously on NFS1 and ISCU, the authors a priori rule out double 'artificial' persulfuration. However, the cryo-EM grid preparation was carried out on a timescale of about 1 minute. According to the data presented by the authors using cysteine labeling, within this timeframe, it is possible to have more than one turnover per NIAUX complex molecule. Therefore, it is impossible to exclude that the observed structure with two persulfides reflects these two successive reactions (since this structure was obtained from a subset or particles 'selected' on the grid and therefore does not correspond to 100% of the sample in solution). Please consider addressing this point in the final version of the manuscript.

The authors assume that the studies presented in this manuscript are necessarily 'physiological,' contrary to everything that has been described previously in the literature. Specifically, at line 241, the authors assert that their iron-loaded complex is physiological. However, these studies, like the previous ones, are conducted in solution with overproduced and purified proteins, and there is no evidence that the iron binding mode they report in their NIAU complex is indeed the same in a cell, especially considering that the iron donor is unknown, and the sample was prepared with a 5X excess of ferrous ions. Please consider this point as well in the final version of the manuscript.

Aside from these few points, the reviewer can only acknowledge the very high quality of the structural work and the significance of the results presented in unveiling the initial steps of the mechanism of iron-sulfur cluster assembly at the molecular level. Therefore, the reviewer highly recommends the publication of this manuscript in Nature Communications after the suggested corrections have been made.

Reviewer #4 (Remarks to the Author):

Manuscript: NCOMMS-23-53565-T

Title: " Mechanism and structural dynamics of sulfur transfer during de novo [2Fe-2S] cluster assembly on ISCU2"

Author(s): Vinzent Schulz, Ralf Steinhilper, Jonathan Oltmanns, Sven-A. Freibert, Nils Krapoth, Uwe Linne, Sonja Welsch, Volker Schünemann, Bonnie J. Murphy, Roland Lill

Comments to the authors:

The authors of the present manuscript entitled “Mechanism and structural dynamics of sulfur transfer during de novo [2Fe-2S] cluster assembly on ISCU2” by Vinzent Schulz et al., deals with the assembly of the iron-sulfur cluster in eukaryotes by the ISC machinery. Biogenesis of Iron-Sulfur Cluster (Fe-S) is an essential cellular process in biology that deserves to be studied at a molecular level. Indeed, defects in this process are associated to numerous diseases, including myopathy, ataxia,...

In contrast to bacteria, in eukaryotes there are two machineries dedicated to Fe-S assembly, the ISC machinery found in the mitochondria and that requires numerous proteins; and the CIA machinery presents in the cytosol that contains less proteins and dedicated to the maturation of cytosolic and nuclear Fe-S proteins. Both machineries involve a scaffold platform whose function is to assemble an Fe-S cluster (step 1) before transfer to target proteins (step 2). The present manuscript deals with the step 1 of the core-ISC mitochondrial machinery involving ISCU2 (the Fe-S scaffold, U), NFS1-ISD11-ACP (cysteine desulfurase, NIA), FXN (frataxin, X) and FDX2 (ferredoxin, electron donor). There are several structures in the literature (X-ray or Cryo-EM) of the core-ISC complex, +/- X protein, +/- Fe, +/- Zn as metal bound to ISCU active site showing differences in metal coordination on ISCU suggesting a structural plasticity of ISCU assembly site. Here, the authors, using a combination of structural biology (Cryo-EM), biochemical and Mössbauer spectroscopy under anaerobiosis, elucidated the Fe-dependent mechanism of persulfide transfer from cysteine desulfurase NFS1 to ISCU2 by characterizing various intermediates of this multi-stage process. High-resolution cryo-EM structures obtained from anaerobically prepared samples provided snapshots that both visualized different stages of persulfide transfer from Cys381NFS1 to Cys138ISCU2 and clarified the molecular role of frataxin in optimally positioning assembly site residues for sulfur transfer. Biochemical analyses assigned ISCU2 residues essential for sulfur transfer, and revealed that Cys138ISCU2 received the persulfide from Cys381NFS1 without Cys95ISCU2 as intermediate. Mössbauer spectroscopy allowed to define the dynamic changes in Fe coordination during the various reactions.

The present work provides a highly detailed molecular understanding of both the mechanism and structural dynamics of sulfur transfer within the core ISC complex. Further, it confirms the molecular role of FXN in this process, facilitating sulfur transfer by rearrangement of the ISCU2 assembly site.

The paper is well-written and very clear, taking you step-by-step towards an understanding of the mechanism of Fe-S biogenesis on ISCU2. The results are then summarized in a well-illustrated figure (Figure 5). On the whole, this is an exceptional and rigorous study that deserves to be published in Nature

Few comments:

1/ page 12 (line 373). It's not clear why histidine 137 from ISCU is proposed as ligand to Fe-bound ISCU when NIA binds ISCU, while Cryo-EM clearly shows that this residue does not. Furthermore, figure 5 clearly shows that histidine 137 no longer binds iron when NIA arrives.

2/ Figure 5: not clear how step 2 to step 3 occurs. How is it possible that in the presence of L-cysteine substrate, the Cys381NFS1 that coordinates Fe ion on ISCU2 becomes persulfidated ? (this implies that it decoordinates Fe at a given time) from a chemical point of view this is not obvious. Please clarify that point.

Response to Reviewers' comments

Manuscript NCOMMS-23-53565-T, Schulz, Steinhilper et al.

We thank the Reviewers for their careful reading of our manuscript and the very positive evaluation. We have addressed all concerns of the Reviewers in our revised version and in the rebuttal letter below. We also have polished the text of our manuscript one more time, slightly corrected Table 3, and adopted the formatting of the manuscript to the standard of *Nature Communications*. We hope that our manuscript will now be acceptable for publication in *Nature Communications*.

Note: Reviewers' original text is in italics; our response is in plain text. The changes in our manuscript during revision are highlighted in the file Schulz.revision-changes.pdf to ease the reviewing process.

Referee #1 (Remarks to the Author):

..... (first paragraphs deleted)

The work presented here is of high significance in the field, considering that is an extra step in the direction of understanding more clearly, what the frataxin role is.

The methodology used is sound and the conclusions made are justified by the data. I admire the clever use of persulfide transfer reactions labelled with thiol-specific iodoacetyl Tandem 746 Mass Tag™ (iodoTMT) reagents for the HRMS experiments.

Several minor points:

1. In the Experiment "Persulfidation assay with iodoTMT-based persulfide quantification", what will happened if instead of "100 – 150 μM FeCl₂", the authors would add ZnCl₂?

We have not performed such an experiment, because in principle it has been done earlier by Dr. D'Autreaux's group, even though under different (aerobic) conditions and using a different alkylation reagent. This work showed qualitatively similar results, yet at slower transfer rates and with more side reactions. At any rate, the use of Zn²⁺-bound ISCU2 is non-physiological, because Zn²⁺ will allow persulfide transfer (as shown earlier), but then is unable to support the synthesis of the cluster in the FDX2-catalyzed step. The progress of our study is that we now can analyze the function of the core ISC system under close-to-physiological conditions.

2. More details would be helpful to replicate the LC-MS methodology for the source parameters for both Orbitraps.

We have expanded the text in the Methods section to better explain these details.

3. What was the amino acids sequence of the frataxin used in the experiments?

Supplementary Table 3 contains the information for all plasmids (and hence the protein sequences) used in our study. To make this more clear, we have added a text to Supplementary Table 3 to read "Plasmid constructs used in this study are based on the canonical sequences from the Uniprot database (uniprot.org)." As requested by the Reviewer, Supplementary Table 3 also refers to the published sequence of frataxin (Freibert et al., 2021 *Nature Comms.* 12:6902, Supplementary Table 4).

4. Can the MS2 spectra be added in the extended figures?

As requested, we have added a Supplementary Fig. 13 showing exemplary ion traces.

5. I recommend one more reading for some words that used a synonym instead of the regular English term

We checked our manuscript carefully, and amended the text where necessary.

Reviewer #2 (Remarks to the Author)

..... (first paragraph deleted)

That said, I have a few relatively minor comments and questions.

1. Given the delicate nature of the sulfur-based structures to be resolved in this study, is there a reason the authors elected to use a fairly high total electron dose (60 e-/Å²) during data collection? This might be expected to run the risk of breaking relatively labile S-S bonds.

The rationale for using a relatively high electron dose of 62 e-/Å² during cryo-EM data collection is to improve image contrast, which can be advantageous for image alignment and 3D classification of subtle structural differences; because data are acquired in movie mode and dose-weighting is applied during image processing, high-resolution information contained in the earlier frames of the acquisition is preserved. The dose-weighting procedure applied in our initial analysis (first during motion correction, using a generic model of dose-dependent damage (Grant & Grigorieff 2015; DOI: 10.7554/eLife.06980), and then empirically using dataset-derived b-factors during Bayesian polishing (Zivanov et al 2018; DOI: 10.1107/S205225251801463X) down-weights high-frequency information from later frames of the acquisition, thereby limiting the contribution of dose-damaged information to the final reconstruction.

To confirm that the result is not due to dose damage, we have carried out additional analyses, truncating movies to include only the early frames of the acquisition at the stage of particle polishing. At a total dose of 2 e-/Å², which is lower than that shown by Kato et al 2021 (DOI: 10.1038/s42003-021-01919-3) to preserve disulfide bonds (3.3-5 e-/Å²), we obtain a similar reconstruction, consistent with our initial analysis, albeit at slightly lower resolution (due to the fact that each low-dose particle image is inherently lower-SNR than the corresponding dose-weighted, full-frame particle image). We now present this issue in the additional Supplementary Fig. 4 showing the results of our analysis.

2. How do the authors account for the observed structural differences between their Fe-bound structures and previously published Zn-bound structures? Do the varying atomic distances reflect different transient intermediates of the complex during the reaction or do the authors think the presence of Zn²⁺ may introduce structural artifacts?

Higher resolution structures allow more confident modelling of metal coordination. It seems clear that the ISCU2 metal site is highly dynamic, and it therefore seems possible that changing the identity of the metal ion could affect the coordination; however, as the Zn²⁺-inhibited complex is not physiologically relevant, we prefer not to speculate in the manuscript on potential functional implications of the slightly different metal coordination.

3. *Sample and reaction preparation were performed under anaerobic conditions. Was sample vitrification similarly maintained in anaerobic conditions?*

Yes, the entire sample preparation process including sample vitrification was performed under anaerobic conditions. The Vitrobot, which was used for cryo-EM sample vitrification, was located inside an anaerobic tent. We have updated the relevant Methods section to make this point clear.

4. *What was the reason for the 1.2x downsampling of the particles in the final reconstructions?*

Downsampling is commonly applied in cryo-EM data processing for efficient use of computational resources. 1.2x downsampling of the particles results in a pixel size of 1.03015 Å, which corresponds to a Nyquist frequency of 2.0603 Å. Thus, the highest possible resolution that could be obtained in the final reconstruction of these particles is 2.06 Å. As none of the maps reached this resolution (the highest local resolution in the consensus map is 2.29 Å) there is no expected negative influence of a 1.2x downsampling for the final reconstructions.

5. *In figure 1, it would be nice to see a more convincing representation of the map density that provided evidence for persulfidated cysteines.*

We have improved Fig. 1 to meet the Reviewer's request.

Reviewer #3 (Remarks to the Author)

..... (first and part of second paragraph deleted)

..... *The reviewer is not an expert in Mossbauer spectroscopy and cannot truly assess the quality of the interpretation. However, the authors seem to assume that the signals observed by this method necessarily correspond to the species observed by cryo-electron microscopy. For instance, the authors consider that the signal they term 'component 2' is necessarily associated with the structure observed by cryo-EM. However, the sample concentrations between the two types of experiments are quite different, and one cannot exclude that at such high concentrations used for the Mossbauer spectroscopy study, a significant amount of iron could be released, and the signal corresponding to an isomer shift of 1.1 – 1.15 mm/s could be that of this free iron.*

We do not claim in our manuscript that the “*signals observed by this method* (i.e. Mössbauer) **necessarily** correspond to the species observed by cryo-electron microscopy”. However, the equilibrium between two states seen in Mössbauer spectroscopy correlates nicely with the structural changes seen by cryo-EM. As stated in our manuscript, the two techniques not necessarily see the exactly same state, which is not surprising due to the different conditions that could slightly change the malleable cluster assembly site of ISCU2. Further, we admittedly do not understand the Reviewer's view why “*a significant amount of iron could be released*” at high protein concentrations. Under such conditions, the binding of iron (particularly to low-affinity sites) would in fact be increased. For such Fe (II) nonspecifically bound to protein, we would expect δ -values at 1.2-1.3 mms^{-1} for sixfold N/O octahedrally coordinated Fe(II). This is indeed the case for $\text{Fe}+(\text{NIAx})_2$ (Fig. 4e, Tab. 3) but not for the other ISCU2-containing samples. Since we do not observe sixfold N/O coordinated iron (II) in the samples containing ISCU2 we consider the presence of adventitious bound

iron as unlikely (as already stated in our manuscript).

Minor points:

Regarding the observation (or at least the presentation) of a structure of the NIAUX complex with two persulfides simultaneously on NFS1 and ISCU, the authors a priori rule out double 'artifactual' persulfuration. However, the cryo-EM grid preparation was carried out on a timescale of about 1 minute. According to the data presented by the authors using cysteine labeling, within this timeframe, it is possible to have more than one turnover per NIAUX complex molecule. Therefore, it is impossible to exclude that the observed structure with two persulfides reflects these two successive reactions (since this structure was obtained from a subset of particles 'selected' on the grid and therefore does not correspond to 100% of the sample in solution). Please consider addressing this point in the final version of the manuscript.

We fully agree with the Reviewer that a second round of sulfur release from cysteine could have happened leading to two persulfides (despite the fact that our cryo-grid solutions were prepared on ice and blotted at 4°C, implying a rather slow reaction). This interpretation is (possibly a bit cryptically) stated in our manuscript as “*This arrangement could either represent both pre- and post-sulfur-transfer states*”. However, since this view is speculative, we also offered an alternative interpretation as “*or be explained by a mixture of particles in either pre- or post-sulfur transfer states*”. As a reaction to the Reviewer’s valid concern, we have better explained the possibility of a second round of sulfur release from Cys by NFS1 leading to the two persulfides (addition of “*due to a second round of persulfidation*”).

The authors assume that the studies presented in this manuscript are necessarily 'physiological,' contrary to everything that has been described previously in the literature. Specifically, at line 241, the authors assert that their iron-loaded complex is physiological. However, these studies, like the previous ones, are conducted in solution with overproduced and purified proteins, and there is no evidence that the iron binding mode they report in their NIAU complex is indeed the same in a cell, especially considering that the iron donor is unknown, and the sample was prepared with a 5X excess of ferrous ions. Please consider this point as well in the final version of the manuscript.

The Reviewer is right that biochemical experiments with isolated systems are *per se* not truly “physiological”. In our manuscript, we do not claim this. However, our synthesis reaction i) contains all the compounds known to be involved *in vivo* for the studied steps, ii) was done under anaerobic conditions (to avoid artefacts), and iii) occurred at rapid rates (much faster than previously reported). Based on these points, we believe that our system is as close as currently possible to physiological conditions. To meet the Reviewer’s concern, we have deleted the term “physiological” at line 241 and at two positions in Introduction.

Aside from these few points, the reviewer can only acknowledge the very high quality of the structural work and the significance of the results presented in unveiling the initial steps of the mechanism of iron-sulfur cluster assembly at the molecular level. Therefore, the reviewer highly recommends the publication of this manuscript in Nature Communications after the suggested corrections have been made.

We thank the Reviewer very much.

Reviewer #4 (Remarks to the Author)

..... (first two paragraph deleted)

The paper is well-written and very clear, taking you step-by-step towards an understanding of the mechanism of Fe-S biogenesis on ISCU2. The results are then summarized in a well-illustrated figure (Figure 5). On the whole, this is an exceptional and rigorous study that deserves to be published in Nature

We thank the Reviewer for the enthusiastic evaluation.

Few comments:

1/ page 12 (line 373). It's not clear why histidine 137 from ISCU is proposed as ligand to Fe-bound ISCU when NIA binds ISCU, while Cryo-EM clearly shows that this residue does not. Furthermore, figure 5 clearly shows that histidine 137 no longer binds iron when NIA arrives.

We thank the Reviewer for spotting this point. We admit that the criticized sentence was unclear and in fact confusing. We had tried to also report the coordination differences in the Zn- and Fe-bound structures, but this led to a possible mis-understanding. In our Discussion, this point is correctly (and hopefully clearly) explained (see Fig. 5). We have re-written (shortened) the criticized sentence in Results.

2/ Figure 5: not clear how step 2 to step 3 occurs. How is it possible that in the presence of L-cysteine substrate, the Cys381NFS1 that coordinates Fe ion on ISCU2 becomes persulfidated ? (this implies that it decoordinates Fe at a given time) from a chemical point of view this is not obvious. Please clarify that point.

Also this point is well-taken. Based on the fast rates of NFS1 persulfidation (see Supplementary Fig. 11d) it is well possible that *in vivo* the (NIA)₂ complex binds with an already persulfidated NFS1 to Fe-ISCU2 thus immediately creating stage 3 of Fig. 5 (without an intermediate stage 2). From our experiments, we know, however, that stage 2, upon addition of Cys, can readily be converted to stage 3, possibly due to the high dynamics of the Cys-loop of NFS1. We therefore decided to present the model as in Fig. 5 because this reflects the coordination observed in our cryo-EM structures. Of course, without measuring the relative rates of persulfidation (forming (N^SIA)₂ and Fe-ISCU2 binding to (NIA)₂ (biochemically a quite demanding task), it is impossible to say which pathway is the likely *in vivo* situation. To clarify this point, we have re-written the relevant paragraph of Discussion to point out to the reader that the *in vivo* pathway may be different from the stages shown in Fig. 5.

REVIEWERS' COMMENTS

Reviewer #3 (Remarks to the Author):

The authors have now addressed all of my previous remarks and I have no further comments. I support publication of this manuscript in Nature communication without further modification.

Reviewer #4 (Remarks to the Author):

Manuscript: NCOMMS-23-53565-A

Title: " Mechanism and structural dynamics of sulfur transfer during de novo [2Fe-2S] cluster assembly on ISCU2"

Author(s): Vinzent Schulz, Ralf Steinhilper, Jonathan Oltmanns, Sven-A. Freibert, Nils Krapoth, Uwe Linne, Sonja Welsch, Volker Schünemann, Bonnie J. Murphy, Roland Lill

Comments to the authors:

The reviewer is satisfied by the answers of the authors to his/her questions and also by the responses to the points raised by the other reviewers. The revised manuscript is ready for publication to Nature Communications!